# Exploring patterns in precipitation intensity–duration–area–frequency relationships using weather radar data

**Talia Rosin[1], Francesco Marra[2], and Efrat Morin[1]**

[1]The Fredy and Nadine Herrmann Institute of Earth Sciences, the Hebrew University
of Jerusalem, Jerusalem, 9190401, Israel
[2]Department of Geosciences, University of Padua, Padua, Italy

**Correspondence:** Talia Rosin (talia.rosin@mail.huji.ac.il) and Efrat Morin (efrat.morin@mail.huji.ac.il)

**Abstract.** Accurate estimations of extreme precipitation return levels are critical for many hydrological applications. Extreme precipitation is highly variable in both space and time; therefore, to better understand and manage the related risks, knowledge of their probability at different spatial–temporal scales is crucial. We employ a novel non-asymptotic framework to estimate extreme return levels (up to 100 years) at multiple spatial–temporal scales from weather radar precipitation estimates. The approach reduces uncertainties and enables the use of relatively short archives typical of weather radar data (12 years in this case). We focus on the eastern Mediterranean, an area of high interest due to its sharp climatic gradient, containing Mediterranean, semi-arid, and arid areas across a few tens of kilometres, and its susceptibility to flash flood. At-site intensity–duration–area–frequency relations are derived from radar precipitation data at various scales (10 min–24 h, 0.25–500 km$^2$) across the study area, using ellipses of varying axes and orientations to account for the spatial component of storms.

We evaluate our analysis using daily rain gauge data over areas for which sufficiently dense gauge networks are available. We show that extreme return levels derived from radar precipitation data for 24 h and 100 km$^2$ are generally comparable to those derived from averaging daily rain gauge data over a similar areal scale. We then analyse differences in multi-scale extreme precipitation over coastal, mountainous, and desert regions. Our study reveals that the power-law scaling relationship between precipitation and duration (simple scaling) weakens for increasing area sizes. This finding has

implications for temporal downscaling. Additionally, precipitation intensity varies significantly for different area sizes at short durations but becomes more similar at long durations, suggesting that, in the region, areal reduction factors may not be necessary for computing return levels over long durations. Furthermore, the reverse orographic effect, which causes decreased precipitation for hourly and sub-hourly durations, diminishes for larger areas. Finally, we discuss the effects of orography and coastline proximity on extreme precipitation intensity over different spatial–temporal scales.

## 1 Introduction

Extreme precipitation is the main trigger of hazards such as floods and landslides that have severe impacts on human beings and livelihoods, causing environmental, societal, and economic damage worldwide – including loss of life (Barredo, 2009; Borga and Morin, 2014). Extreme precipitation is highly variable in both space and time, as various physical processes are involved in its generation. Knowledge about the spatial–temporal scales at which extreme precipitation interacts with catchments, and of the probability of occurrence of extreme precipitation at such scales, is thus crucial for infrastructure design, as well as an improved understanding and management of the related risks and impacts of floods on ecosystems and communities (Mascaro et al., 2023; Wright et al., 2017; Peleg et al., 2018; Mélèse et al., 2019).

Extreme precipitation frequency has traditionally been estimated using techniques based on the extreme value theory. These methods focus on independent and identically distributed random variables and use the maxima within a temporal block (commonly 1 year) assuming an infinite number of events $n$ is observed in each block ($n \to \infty$). Alternatively, the peaks exceeding an asymptotically high threshold $\theta$ ($\theta \to \infty$) can be used. In these conditions, the cumulative distribution function of the block maxima can only converge to a generalised extreme value (GEV) distribution. These asymptotic techniques use a limited fraction of the available data (only the block maxima or the values exceeding a very high threshold) and thus require long datasets in order to provide accurate estimates. They are typically limited to applications on rain gauge data, which generally possess the longest datasets. Rain gauge networks, however, are often sparse worldwide (Kidd et al., 2017), making a complete and adequate statistical characterisation of extreme precipitation difficult. In general, these methods are prone to large uncertainties when estimating return periods longer than the available record.

Recently, non-asymptotic approaches such as the Metastatistical Extreme Value (MEV) approach (Marani and Ignaccolo, 2015) and the simplified MEV (SMEV) approach (Marra et al., 2019a) have been proposed (Marani and Ignaccolo, 2015; Marra et al., 2018; Vidrio-Sahagún and He, 2022; Zorzetto et al., 2016). Unlike traditional methods, these approaches derive an extreme value distribution based on the bulk distributions of so-called "ordinary" events rather than only considering the extremes. To this end, the ordinary events are defined as all the independent realisations of the variable of interest (Zorzetto et al., 2016). The advantage therefore is that these methods use a large proportion of the available data. This significantly decreases the uncertainty of the estimated parameters, allowing for a more accurate estimation of rare return levels from short records and records containing measurement errors affecting the extremes (Marani and Ignaccolo, 2015; Marra et al., 2018; Zorzetto et al., 2016).

Non-asymptotic techniques are extremely well suited to radar precipitation data, which are increasingly available in a systematic manner and are considered appropriate to capture the spatial variability of extreme precipitation, including events with limited spatial extent (Lengfeld et al., 2020; Pöschmann et al., 2021). Weather radars provide fine-scale precipitation data at high spatial and temporal scales over both land and sea. Thus, applying these alternate approaches to radar data provides a significant opportunity, not only to estimate return levels in areas where point precipitation data are unavailable but also to incorporate the areal component in extreme precipitation analysis – a natural advantage of using radar data. It should be noted however that radar products do have well-documented issues with precipitation estimation errors, which translate to the extremes, and must be considered (Peleg et al., 2018; Marra et al., 2019b).

The MEV and SMEV frameworks have been successfully applied to both point and spatial precipitation and to a variety of locations. They are able to estimate return levels corresponding to return periods much longer than the record length better than traditional methods (Hu et al., 2020; Marra et al., 2018; Schellander et al., 2019; Zorzetto et al., 2016). These approaches therefore offer a promising means of managing the risks associated with natural hazards by estimating the frequency of extreme precipitation events at multiple scales from radar data. In particular, the SMEV approach has been applied to several case studies, including this study location, and demonstrated to have a number of advantages: it is less sensitive to measurement errors typical of radar estimates (Marra et al., 2018) and to the use of short records (Marra et al., 2018; Hu et al., 2020; Zorzetto et al., 2016) as it correctly represents the tail of sub-daily precipitation intensities (Wang et al., 2020; Marra et al., 2020). Comparing the SMEV to asymptotic models based on the GEV distribution, Vidrio-Sahagún and He (2022) found that SMEV-based models demonstrated superiority in the analysis of non-stationary time series due to their higher accuracy, equivalent or better fitting efficiency, as well as lower uncertainty compared to other tested models, including the MEV.

The statistical characteristics of extreme precipitation are commonly quantified using intensity–duration–frequency (IDF) curves, which are cumulative distribution functions of annual precipitation maxima conditioned on duration. IDF curves are used to derive design storms and are employed in hydrological design and as decision support information in flood risk and water management. IDF curves display precipitation at the point scale for a specific location and are generally computed from rain gauge data. However, IDF curves clearly do not address the aspect of spatial precipitation. Areal precipitation intensity is generally computed by multiplying point precipitation by an areal reduction factor (ARF) (Panthou et al., 2014). The ARF is a corrective coefficient, which is defined as the ratio between the areal average precipitation and point precipitation. ARFs can be computed using either design precipitation intensity data or actual precipitation intensity, depending on the calculation method used, the available information, and the purpose of the analysis. In this study we refer to ARFs generally in the context of design precipitation. Typically, ARFs are presented as a set of curves showing the variation of ARF with precipitation intensity, duration, and frequency (Kao et al., 2020; Sivapalan and Blöschl, 1998; Thorndahl et al., 2019).

Many methods to compute ARFs have been proposed in the literature, including analytical formulations and empirical approaches based on precipitation observations. A comprehensive list can be found in Olivera et al. (2008) and Svensson and Jones (2010). ARF values vary significantly due to a variety of factors such as precipitation characteristics and patterns (e.g. convective or frontal precipitation), location, and surface characteristics (such as topography and altitude) and consequently are only representative of a limited

area around a point. Thus, they must be calculated specifically for each location.

An alternative approach is to extend the concept of IDF curves to also include the areal component and create intensity–duration–area–frequency (IDAF) curves (De Michele et al., 2002; Mélèse et al., 2019). IDAF curves are cumulative distribution functions of precipitation intensity conditioned on duration and area and thus incorporate the variability of precipitation intensity over a range of spatial scales. They are becoming increasingly popular as they are more useful when storm severity needs to be characterised over an area, say a catchment, for example when designing hydraulic structures. To estimate IDAF curves for an area using rain gauge measurements, point rainfall return levels are generally transformed to areal rainfall return levels using ARFs. This requires a high-density network of gauges, and, therefore, issues arise in the light of the quality of hydro-meteorological measurements, especially in developing countries. However, with remote sensing data, the calculation of ARFs is unnecessary as the use of distributed precipitation allows for the direct estimation of areal precipitation intensity.

In this study we apply the SMEV framework to examine extreme precipitation at various spatial scales for the first time, in order to investigate the impact of area size on local extremes. While the SMEV framework has demonstrated efficacy in successfully estimating extreme rainfall, its prior applications have all been confined to either the point-scale (in the case of rain gauge data) or pixel-scale (when utilising radar rainfall data) analyses at different temporal scales. Here we extend the application of the SMEV to estimate extreme return levels up to 100 years across multiple spatial and temporal scales. The analysis is centred on a 12-year radar precipitation dataset covering the eastern Mediterranean. We compare extreme return levels derived from radar precipitation data to those derived from averaging rain gauge data in several areas in which a sufficient rain gauge density is available. We then construct IDAF curves to investigate the characteristics of extreme precipitation in coastal, desert, and mountainous regions and evaluate them over different spatial and temporal scales.

## 2 Study area and precipitation data

This study focuses on the eastern Mediterranean, an area of high interest due to its sharp climatic spatial gradient, which ranges from Mediterranean to semiarid and arid across a few tens of kilometres (Fig. 1). This results in catchments with highly non-homogeneous climatic and hydrological conditions (Zoccatelli et al., 2019). Precipitation emerges mainly from cold fronts and the air masses that follow these fronts, which are associated with mid-latitude cyclones during their eastward passage over the eastern Mediterranean (Goldreich, 2003). Additionally, precipitation can be caused by other systems which bring precipitation of a more local nature (Armon et al., 2019). Precipitation occurs primarily during the winter months, with almost no precipitation from June to September.

### 2.1 Radar data

Weather radar data were provided by the Israel Meteorological Service (IMS) from the C-band weather radar located at Beit Dagan, Israel (Fig. 1). Precipitation data cover the periods from the hydrological year 2007–2008 to 2017–2018 (hydrological year is defined here as 1 September to 31 August). As the instrument was sometimes turned off, the archive cannot be considered complete. The data have a temporal resolution of 10 min and, after processing (see below), are converted to a Cartesian grid with a spatial resolution of $500 \times 500 \, \text{m}^2$.

A combination of physics-based corrections and empirical adjustments, optimised for long-term radar archives and high-intensity convective storms, was applied to the radar archive to obtain a high-quality, homogeneous dataset. These corrections consisted of the removal of non-precipitating echoes (i.e. ground clutters) (see Marra and Morin, 2018), correction of beam blockage due to orography (Marra et al., 2014), identification and correction of non-orographic blockages (Marra et al., 2022), correction of beam attenuation during heavy precipitation (Marra and Morin, 2015), and correction of vertical variations in reflectivity (Marra and Morin, 2015; Morin and Gabella, 2007).

After these corrections were performed, the precipitation intensity $R$ ($\text{mm h}^{-1}$) was computed from radar reflectivity $Z$ ($\text{mm}^6 \, \text{m}^{-3}$) using a fixed power-law relationship in the following form:

$$Z = 3.16R^{1.5}, \tag{1}$$

which is well suited for the convective precipitation of the region (Morin and Gabella, 2007).

Precipitation intensity at the ground was calculated by taking the average of the two highest intensities along the vertical dimension for elevation angles up to 5° and converted to Cartesian coordinates (Marra et al., 2022). The final radar archive was obtained after a two-step bias adjustment based on daily rain gauge archive data (Morin and Gabella, 2007; Marra and Morin, 2015). This adjustment aimed at optimising the bias and dispersion of rainfall depths during independent meteorological events. A full description of the radar data elaboration procedure and overall data quality is provided in Marra et al. (2022), who demonstrated that the use of these corrections significantly improves the quality of the radar precipitation data archive. Note however, that there are still some estimation errors in the data, due to issues such as the underestimation caused by range effects (visible in the northern and southern portions of the domain for areas farther than approximately 100 km from the radar) and the overfilling of blocked beams (Marra et al., 2022).

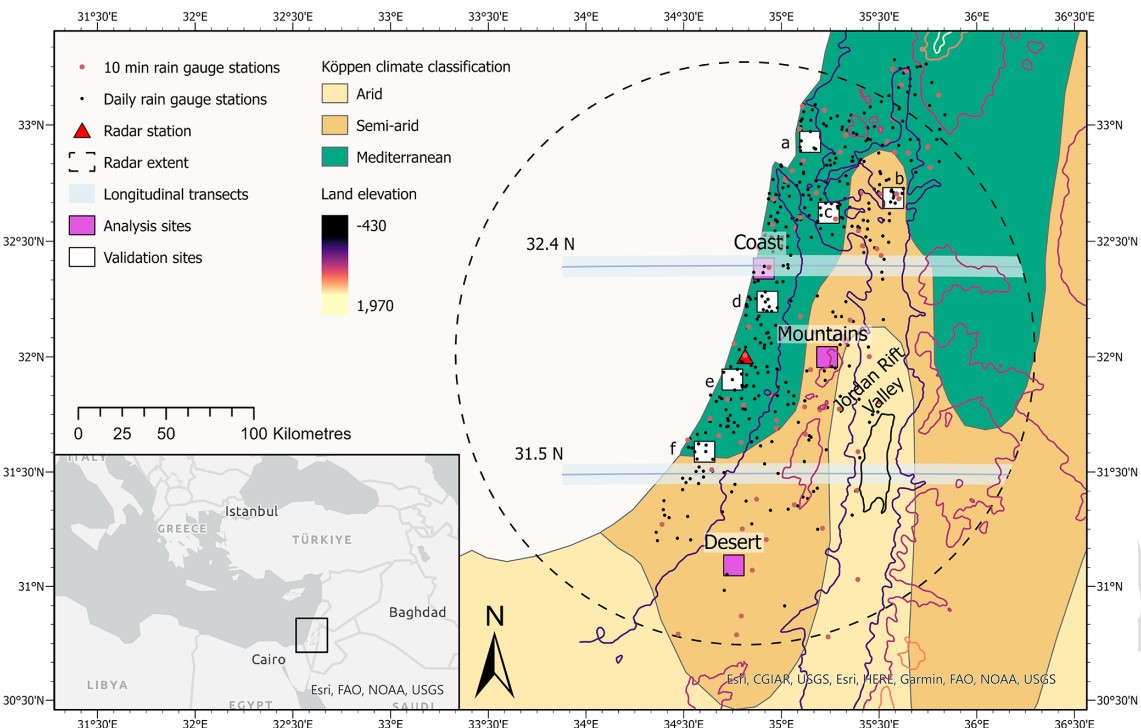

**Figure 1.** Map of the study area displaying terrain elevation, the location of the radar station and the 140 km radar range, the location of 10 min and daily precipitation gauges, the location of the six validation sites analysed in Fig. 3a–e, the three analysis boxes analysed in Figs. 5–7 (labelled Coast, Desert, and Mountains), and the location of the two transects displayed in Fig. 8.

## 2.2 Rain gauge data

Rain gauge data were provided by the IMS and consist of two datasets:

i. A daily archive contains the precipitation depth measured over the 24 h period from 06:00 UTC to 06:00 UTC the following day. This dataset is used to adjust and validate the weather radar archive, to define storms (see Sect. 3.1), and to evaluate IDAF relations at the 24 h, 100 km² scale (see Sect. 3.3).

ii. A 10 min archive from automatic stations contains precipitation intensity data with a 10 min temporal resolution. This dataset is used to adjust the radar-derived statistics at multiple temporal scales (see Sect. 3.2).

The rain gauge datasets are quality-controlled by the IMS. In addition, stations located in regions with low weather radar data quality (e.g. due to residual contamination by ground clutters and blockages) are removed to avoid negative impacts on the bias adjustments. In total, 437 daily stations and 65 10 min stations are included in the analyses (Fig. 1). For the case of the 10 min data used to adjust the radar statistics, hydrological years with more than 10 % missing radar data are removed to ensure accurate quantification of the precipitation statistics, as recommended by Marra et al. (2020).

## 3 Methodology

Extreme precipitation return levels are estimated across the study area using the novel non-asymptotic SMEV framework proposed by Marra et al. (2019a, 2020), a simplified version of the original MEV framework proposed by Marani and Ignaccolo (2015). The MEV and SMEV approaches are based on the concept of "ordinary events", which are all the independent realisations of the process of interest. Unlike classic extreme value theory, which only exploit a small subset of the data, i.e. the annual maxima or the peaks exceeding a high threshold, they make use of a greater proportion of observations to fit the distribution parameters, thus decreasing the parameter estimation uncertainty.

The SMEV is a modified version of the MEV; it neglects the interannual variability of the distribution of ordinary events and their number of yearly occurrences (Marra et al., 2019a). The SMEV formulation significantly reduces the number of parameters and allows for a direct interpretation of their meaning. This results in a simpler formulation for the non-exceedance probabilities of extreme rainfall and more robust parameter estimation. Several studies have applied the SMEV to precipitation frequency analysis over different regions (Marra et al., 2019a, 2020; Miniussi and Marra, 2021; Araujo et al., 2023), including over the study area (Marra et al., 2022), and have demonstrated the robustness of the

method's assumptions and its ability to reproduce extreme frequencies from relatively short records. The SMEV is used here to estimate precipitation events of varying area sizes and durations, so that spatial and temporal effects on extreme precipitation can be analysed.

### 3.1 Identification of the independent storms and of the ordinary events

We follow the unified approach proposed by Marra et al. (2020), in which "storms" are defined as independent meteorological objects, and one "ordinary event" at the scale of interest (both spatial and temporal) is extracted from each storm. Individual storms used for the analysis are first identified at the regional scale (i.e. the entire study area) from the daily rain gauge dataset. A day is considered wet when at least five rain gauges are measuring precipitation greater than 0.1 mm, and a storm consists of consecutive wet days separated by at least 1 dry (i.e. not wet) day. More information is given in Marra et al. (2022). A total of 498 storms were identified in the 12-year dataset. Storms are then identified locally at each pixel within the study area using the radar precipitation data; if any radar precipitation occurs at a pixel (or when considering the areal scale over the selected ellipse centred on that pixel (described below)) during one of the gauge-identified storms it is classed as a storm for that pixel.

Ordinary events at the spatial (area) and temporal (duration) scales of interest are then identified at each radar pixel for each storm, with one ordinary event calculated for each storm. Ordinary events are defined as the storm's maximal space- and time-averaged precipitation intensity for a given area, centred at the pixel, and for a given duration. Area and duration are taken from a combination of different preselected area sizes (pixel scale, 10, 50, 100, 500 km) and durations (10, 30 min, 1, 3, 12, 24 h). It should be noted that for each area considered, the number of ordinary events at each pixel is consistent for all the examined durations. This is due to the unified approach used to define the ordinary events, which goes through the identification of independent "storms" separated by at least 24 dry hours.

For the pixel area size (i.e. $500 \times 500\,\text{m}^2$), a time series of radar precipitation data is first constructed at each radar pixel, at the original temporal resolution of 10 min. The ordinary events are then identified using a moving-time-window approach (window size according to the selected duration and at 10 min time steps) to select the period within the storm which maximises average precipitation intensity for the analysed pixel. This is performed for each of the considered durations. Note that these pixel-scale analyses are analogous to the ones presented in Marra et al. (2022).

For areas larger than the pixel size, we adopt a pixel-centred approach, in which precipitation is characterised as an ellipse having the selected area and centred at the analysed pixel. Storms have frequently been approximated as elliptical shapes (Karklinsky and Morin, 2006; Kim et al., 2019;

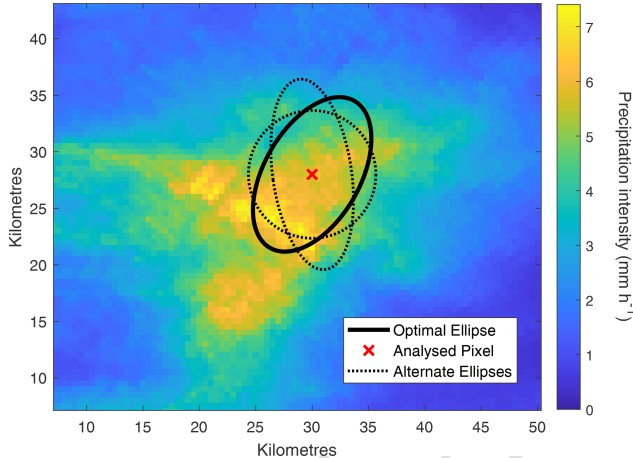

**Figure 2.** Optimum ellipse identification for an area of $500\,\text{km}^2$ and duration of 1 h around an analysed pixel. The selected optimum ellipse and two rejected ellipses are shown.

Northrop, 1998; Olivera et al., 2008) since storms leave long "traces" as they travel, and thus they are better captured by ellipses than circles. Analysing elliptically shaped extreme storms in South Korea, Kim et al. (2019) found that the use of circles versus ellipses resulted in an underestimation of storm-centred ARF values by an average of 20 %, with observed underestimations as high as 70 %.

Ellipses are defined here in terms of ellipticity (i.e. the ratio of the diameter of the minor axis to the major axis) and orientation (the angle formed between the ellipse's major axis and the west–east axis). For each pixel, the best-fitting or "optimum" ellipse for each storm event is identified by systematically varying the ellipticity and orientation, while keeping the ellipse's area fixed, and utilising a moving time window across the selected duration. The combination of ellipticity and orientation which maximises the average precipitation intensity at any period within the storm event is identified as the optimum ellipse best capturing that particular event for the analysed pixel. The process is shown in Fig. 2, which displays the selected optimum ellipse for $500\,\text{km}^2$ area, 1 h duration precipitation, and two rejected ellipses, for an example case.

This ellipse identification process is performed for every combination of area and duration; thus, a total of 24 different ellipses are identified per storm event for an individual pixel (four areas and six durations). Selected ellipses were found generally to have an ellipticity varying between 0.6 and 0.9; this is similar to the results presented by Karklinsky and Morin (2006), who analysed the spatial characteristics of radar-derived convective rain cells over southern Israel, and by Peleg and Morin (2012), who examined convective rain cells over northern Israel. Belachsen et al. (2017) observed slightly lower ellipticities (mean of 0.57) when analysing precipitation characteristics over the Dead Sea area, a region

located in the eastern portion of the domain, forming part of the Jordan Rift Valley (Fig. 1).

Utilising the procedure described above, the ordinary events for different areas and durations are defined across the study area. These ordinary events are then used to calculate return levels, as described in the following section.

## 3.2 Extreme value analysis

Following theoretical analyses (Wilson and Toumi, 2005) and empirical results (Marra et al., 2020, 2023; Zorzetto et al., 2016, among many others), we use a two-parameter Weibull distribution to model the tail of the ordinary events distribution. To define this tail, we left-censor the data (i.e. exclude observations below a selected threshold), following the work performed by Marra et al. (2019a). This analysis demonstrated that low-intensity ordinary events may diverge from the distribution describing the upper tail and thus should not be included when computing parameters describing the upper tail. The left-censoring procedure ignores the intensities of the censored events while still retaining their weight in the probability. The study found that left-censoring values between the 55th quantile and the 80th quantile provide virtually indistinguishable results for the area. Following Marra et al. (2022), here we left-censor the lowest 55 % of the ordinary events. The lower threshold was selected to include the maximum number of ordinary events in the data sample.

Using the two-parameter Weibull model for the left-censored ordinary events, the SMEV cumulative distribution $\zeta$ can be written as

$$\zeta(x) = F(x; \lambda_{D,A,P}, \kappa_{D,A,P})^{n_P}$$

$$= \left[ 1 - e^{-\left(\frac{x}{\lambda_{D,A,P}}\right)^{\kappa_{D,A,P}}} \right]^{n_P}, \qquad (2)$$

where $\zeta$ is the sought yearly non-exceedance probability (e.g. 99 % for the 100-year events); $n$ is the average number of storms per year (and so is the same for all durations for a given area but can change with area size and also varies among pixels); and $\lambda_{D,A,P}$ and $\kappa_{D,A,P}$ are the scale and shape parameters, respectively, which both depend on the examined duration D, area A and pixel P.

The parameter $n$ is computed for each pixel as the total number of storms which are locally wet (identified from the radar precipitation data as described in Sect. 3.1) divided by the number of years in the record. In order to account for possible missing storms in the radar archive, $n$ is adjusted by dividing it by the ratio of the number of regional radar-derived storms compared to the number of gauge-derived storms (Marra et al., 2022).

For each duration and area, the scale and shape parameters are estimated using a least-squares regression in Weibull-transformed coordinates (Marani and Ignaccolo, 2015) at each radar pixel for various combinations of elliptical areas and durations, as described above. The SMEV scale, shape, and $n$ parameters for each duration are next adjusted to incorporate information from the 10 min rain gauge data into the weather-radar-data-derived parameters, using an adjustment procedure developed by Marra et al. (2022). This is performed to mitigate the impact of systematic biases and random errors which have been demonstrated to dominate radar-derived frequency analyses (see Marra and Morin, 2015) and to account for potential missing precipitation events in the radar archive (e.g. when the instrument was turned off). SMEV parameters are estimated for each duration for both the 10 min rain gauge data and the radar data. The local multiplicative biases between the radar-based (pixel size) and the gauge-based parameters are then calculated at each rain gauge location and interpolated using an inverse-distance-weighted method, accounting for both lateral and vertical distances. The radar-derived SMEV scale, shape, and $n$ parameters at each pixel are then adjusted by dividing them by the corresponding interpolated biases, and the adjusted return levels are computed by inversion of the SMEV distribution. This procedure corrects spatial mismatches between the rain gauge and radar data. A full description of this adjustment is presented in Marra et al. (2022).

Note that this adjustment procedure was developed for precipitation at the pixel scale and is applied here to precipitation over areas up to 500 km$^2$. This is necessary as no information on the areal scale can be accurately derived from the rain gauges due to the low density of sub-daily stations. The underlying assumption is that biases in the parameters at the areal scales are similar to biases in the parameters at the pixel scale. We compared radar-derived return level estimates at the 100 km$^2$, 24 h scale against daily rain gauges to get a sense of the accuracy of this assumption (see Sect. 3.3).

The associated uncertainty of the derived return levels is quantified via block bootstrapping with replacement (100 iterations) among the years in the record during the calculation of the return levels, as proposed by Overeem et al. (2008). The technique generates samples by selecting blocks (here a block is defined as a hydrological year) randomly with replacement, so that the number of blocks is the same as in the original record. The ordinary events for each block are concatenated to create the bootstrapped dataset, from which the Weibull parameters and quantiles are estimated, using the procedure described above. This enables the block structure of the original rainfall data to be preserved.

## 3.3 Validation against rain gauge data

Radar precipitation data exhibit various uncertainties which may affect the reliability of the derived return levels. Therefore, the radar-derived return levels are first validated against return level estimates derived from rain gauge data. To extract the areal component from the point-scale rain gauge data, boxes are constructed in locations containing dense networks of rain gauges and the spatial average in these boxes

calculated. Considering the higher density of the daily rain gauge network compared to the 10 min gauges (see Fig. 1), the validation is limited to the duration of 24 h and to areas of $100 \, km^2$, representing a size where boxes with sufficient gauge density can be found. Return levels are calculated from the daily, $100 \, km^2$ gauge-based precipitation spatial averages using the SMEV formulation, utilising the same ordinary events defined for the radar data. IDAF curves are then constructed and compared against radar-precipitation-derived IDAF curves, calculated for 24 h duration and $100 \, km^2$ area size for the central pixel of the $100 \, km^2$ boxes.

As the daily rain gauge data are measured from 06:00 to 06:00 UTC, while the 24 h radar events are defined as the 24 h time window containing the maximal precipitation intensity during each storm, we also convert the radar data to daily time steps, from 06:00 to 06:00 UTC, and calculate $100 \, km^2$, 24 h return levels using the SMEV framework. Computing return levels from these daily 06:00 to 06:00 UTC radar data ensures a direct comparison with the gauge-derived results. The locations of the six analysed sites are displayed in Fig. 1. The $100 \, km^2$ boxes were chosen to contain a minimum of seven rain gauge stations with 30 years of data (1988–2018) at each station, with the gauges evenly distributed inside the box. This is to ensure a reasonable estimate of areal precipitation by spatial averaging and reasonable accuracy in the estimated return levels. Unfortunately, this limits the box locations to coastal and northern lowland areas only, as the mountainous and desert regions contain sparser networks of gauges. Additionally, the locations are required to be at least 1 km away from any 10 min rain gauge stations. This is because data from the 10 min stations are used to adjust the SMEV parameters during the bias correction procedure described in Sect. 3.2. Therefore, using data from daily stations too close to the 10 min stations could affect the independence of the evaluation. The rain gauge data span a 30-year period, in contrast to the 12-year dataset used to derive the radar data results. It was decided to use the whole time series, rather than matching the time periods, so as to produce the most accurate return levels against which to validate the radar-derived results.

There are limitations to this method of comparison, related to the nature of the rain gauge data; the radar-derived results utilise precipitation from optimum ellipses best characterising each individual storm, rather than a fixed square. However, when applied to rain gauge data, the ellipses method becomes problematic, as different rain gauges may be selected for different ellipse options, affecting the spatially averaged precipitation data. Thus, this is the best method available for using rain gauges as a benchmark to evaluate the performance of the radar data and assess the assumptions behind the adjustment procedure.

# 4 Results

Return levels across the study region are calculated using the SMEV framework for varying areas (pixel scale and 10, 50, 100, and $500 \, km^2$) and timescales (10 and 30 min and 1, 3, 12, and 24 h) for a range of return periods (2, 5, 10, 25, 50, and 100 years). The following section presents a comparison between radar- and rain-gauge-derived extreme return levels and then displays maps and IDAF curves of the estimated results.

## 4.1 Validation

Figure 3 presents a comparison between the IDAF curves derived from rain gauge measurements with the 24 h and the daily 06:00 to 06:00 UTC radar data. The associated uncertainty of the results is quantified as the 90 % confidence interval from 100 bootstrap iterations (see Sect. 3.2). The figure demonstrates that both the gauge- and radar-derived results are generally in good agreement, particularly in the case of the 06:00 to 06:00 UTC radar data. This is encouraging as the radar results are computed using only 12 years of data and are adjusted using relations derived for the pixel scale, whilst the gauge results utilise 30 years of data and direct precipitation observations. This validation supports therefore the robustness of the applied framework for the study region, including the use at the areal scale of an adjustment developed for the pixel scale.

Distinctions arise between the 24 h and the 06:00 to 06:00 UTC daily-radar-derived results in certain regions. The 06:00 to 06:00 UTC radar results generally show very similar behaviour to the rain-gauge-derived results for all six sites, as well as similar levels of uncertainty; indeed within the uncertainty intervals, the radar estimates largely cannot be distinguished from the gauge estimates. For sites a, b, c, and d, the 24 h radar data also produce good results, producing return levels very similar to the gauge-derived levels. As expected, the 24 h return levels are higher than the 06:00 to 06:00 UTC radar levels, as the exact time window maximising precipitation intensity for each storm is utilised, rather than the maximal 06:00 to 06:00 UTC period. However, at locations e and f, the radar-derived return levels significantly exceed the rain-gauge-derived levels. Interestingly, this mismatch is specific to these two locations, and the 06:00 to 06:00 UTC radar data yield satisfactory results for these sites.

An analysis by Marra et al. (2022) offers insight into this discrepancy by examining the time of the day at which the highest short-duration intensities (i.e. the ordinary events in the distribution tail, as in this study defined as the largest 45 %) occur over the study area. They found that the highest offshore intensities tend to occur in the early morning (02:00–08:00 UTC) or morning (08:00–14:00 UTC) and then shift to mostly morning (08:00–14:00 UTC) at the coastline and near inland. This is caused by the convergence created by the superposition of the westerly winds typical of Mediter-

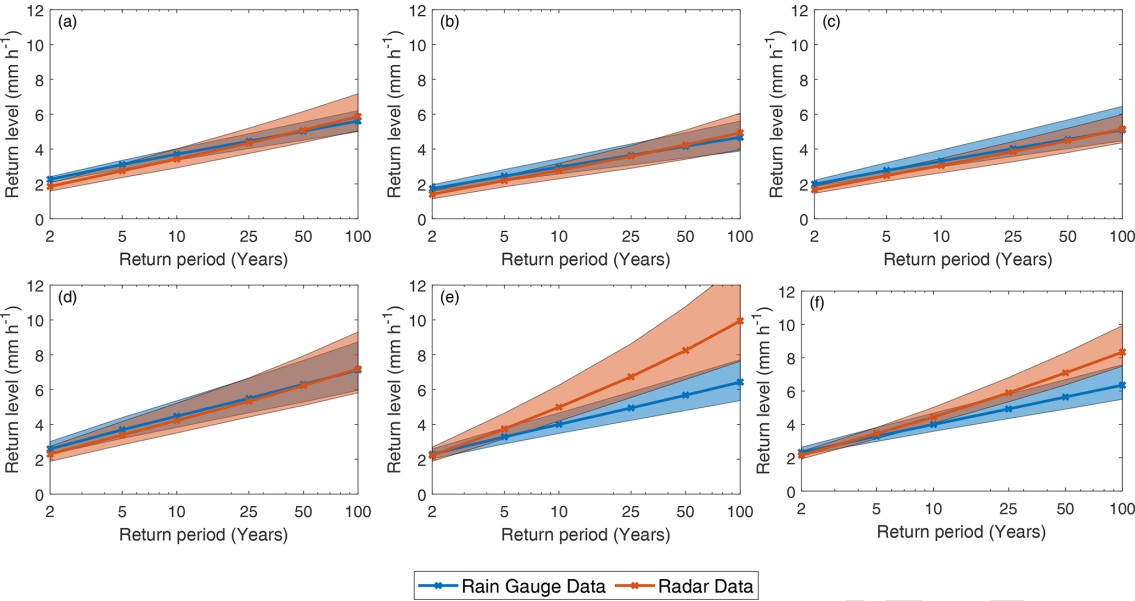

**Figure 3.** Comparison of the $100 \, \text{km}^2$, 24 h precipitation intensity return levels derived from (i) 24 h radar data, (2) daily 06:00 to 06:00 UTC data and (iii) from rain gauge data. The locations of the $100 \, \text{km}^2$ boxes are displayed in Fig. 1.

ranean cyclones with land breeze, which is expected to peak in the early morning hours when the sea is warmest compared to the land. Although Marra et al. (2022) focus only on short-duration rainfall, and the results are given here for 5 24 h events, these findings may still explain the discrepancy between the results. Given that sites e and f are situated on the coastline, high rainfall intensities occur more often in the early mornings between 02:00–08:00 UTC; this could therefore lead to large differences between the maximal 24 h val- 10 ues and the maximal 06:00 to 06:00 UTC event values. Site d is somewhat further east, with Marra et al. (2022) indicating peak rainfall between 06:00 and 08:00 UTC, whilst sites b and c are the most inshore and present high rainfall intensity peak times of 11:00–14:00 UTC and 08:00–11:00 UTC 15 respectively. Therefore limiting the daily data to 06:00 to 06:00 UTC may have a lesser impact on these inland sites.

## 4.2 Return level maps

Figure 4 displays the estimated 25-year return levels for pixel-scale and 10 and $100 \, \text{km}^2$ areas, covering durations of 20 1 h and 24 h. Additional scales are shown in Fig. S1. For shorter durations, the highest return levels are located along the coastline and over the mountainous regions in the north, while the lowest values are found in the desert regions in the south. As expected, increasing the event duration from 25 1 to 24 h results in a decrease in expected precipitation intensity across all area sizes. For long durations, the higher precipitation intensities become concentrated primarily over the central mountain region. Increasing the area size from the pixel scale to $100 \, \text{km}^2$ results in lower return levels but does 30 not significantly alter the spatial distribution of high precip-

itation intensities. An area of very low values, attributed to data quality issues around the radar location, is clearly visible. Immediately south of the radar station, there is also a distinct region of high values. This corresponds to the location of validation site e shown in Fig. 1, which exhibits good 35 agreement between the 06:00 to 06:00 UTC daily-radar-data-derived precipitation intensities and those calculated from gauge data (Fig. 3), suggesting these values are correct. Similarly, the high values along the coastline area are supported by validation sites a, d and f, which also exhibit good agree- 40 ment between the radar- and gauge-derived return levels.

## 4.3 IDAF curves

IDAF relations are used to analyse the effect of duration and area size on extreme return levels. Selected IDAF curves are displayed in Figs. 5–7; the curves are displayed for three dif- 45 ferent regions – the coast, desert, and mountains (see Fig. 1). Return levels over the three regions were computed by calculating the spatial average of all the derived return levels inside a 10 by $10 \, \text{km}^2$ CE1 box. The purpose of displaying return levels for a box rather than a single pixel is to reduce 50 fine noise that may characterise the results. The 90 % confidence intervals are displayed as colour shades, computed via bootstrapping (see Sect. 3.2), with 100 repetitions per radar pixel; thus a total of 10 000 values were calculated for each 10 by $10 \, \text{km}^2$ CE2 box. Results are given for 25-year return 55 periods only, unless otherwise stated, due to space restrictions. Additional return periods are presented in the supplementary material (Figs. S2 and S3), and the corresponding shape and scale parameters for the different regions are presented in Fig. S4. 60

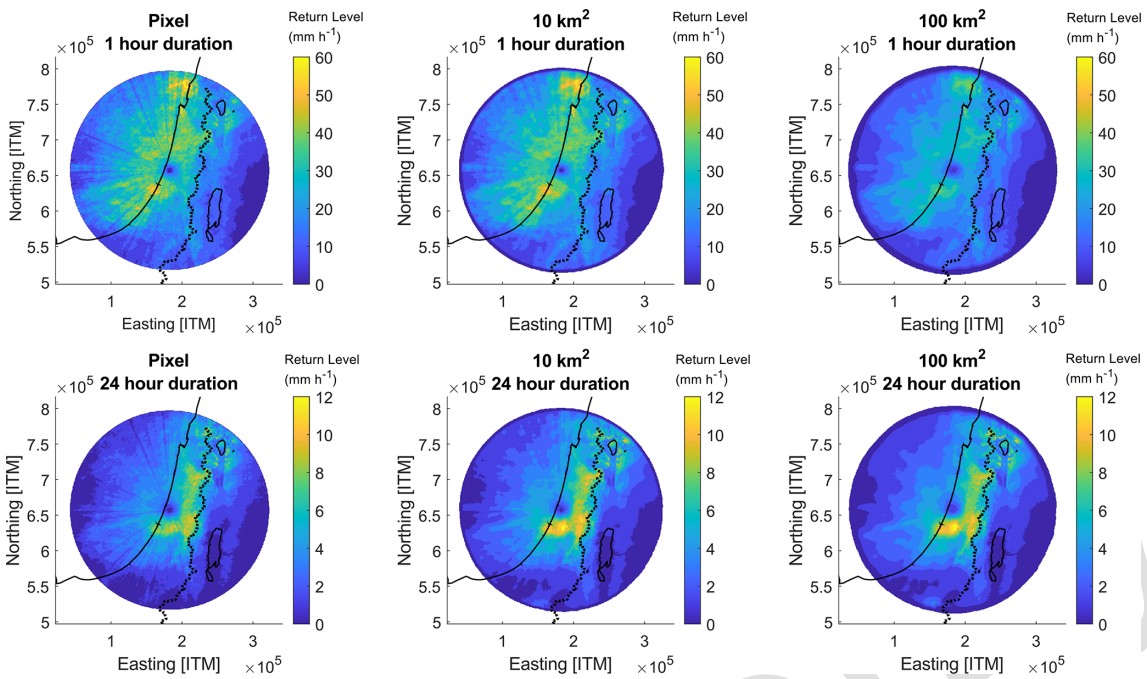

**Figure 4.** Precipitation intensity return levels for varying areas and durations for 25-year return periods. The coastline, the Sea of Galilee in the north, and the Dead Sea in the south are marked by the solid black line. The surface water divide is marked by the dotted black line.

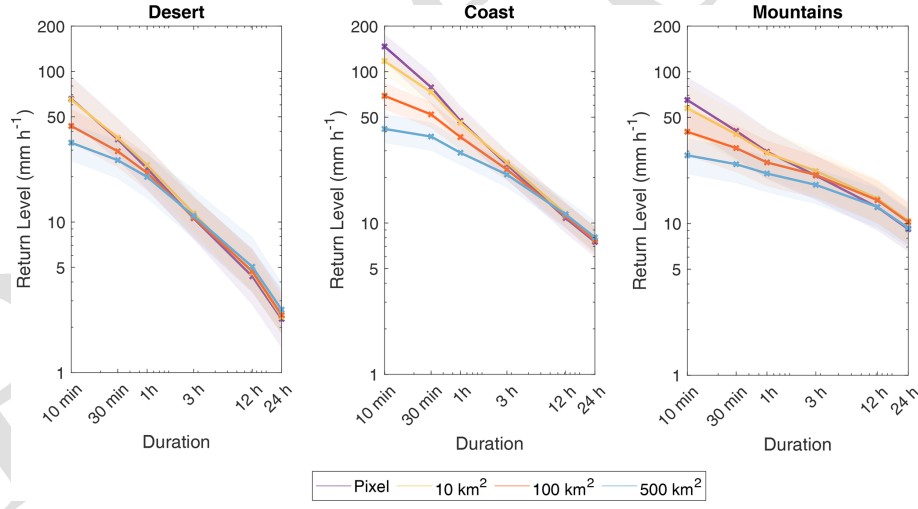

**Figure 5.** IDAF curves estimated for the desert, coast, and mountains for 25-year return periods. Shaded areas represent the 90 % confidence interval from 100 bootstrap samples. The locations of the three analysis sites are displayed in Fig. 1.

The differences in the behaviour of the IDAF curves over the different locations are evident, with higher precipitation intensities over the coastal and mountainous regions, as is expected. Properties of the derived IDAF curves and parameters are discussed in the next section.

## 5 Discussion

The objective of this study is to examine and quantify the effect of area and duration on extreme precipitation statistics using the constructed IDAF curves. Here we identify three main points of interest: (i) the influence of area and duration on precipitation intensity similarity, (ii) the scaling relation between precipitation intensity and duration at different spatial scales, and (iii) the orographic effect. These points are explored in detail below. We then examine the impact of

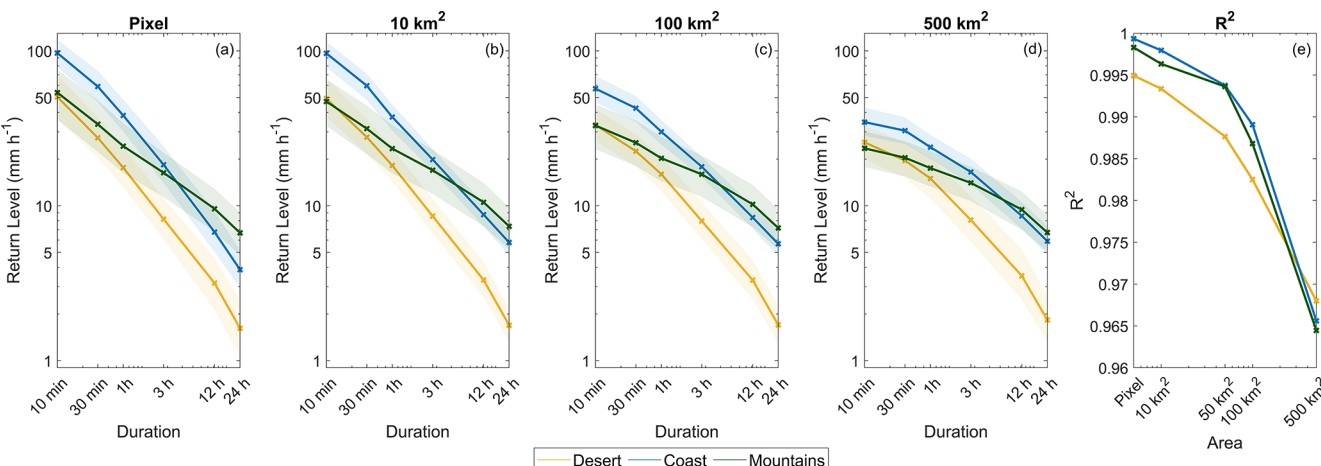

**Figure 6.** CE3 **(a–d)** IDAF curves estimated for different area sizes for 25-year return periods. Shaded areas represent the 90 % confidence interval from 100 bootstrap samples. **(e)** $R^2$ between the log-transformed intensity and duration values presented in **(a–d)** for varying area sizes. The locations of the three analysis sites are displayed in Fig. 1.

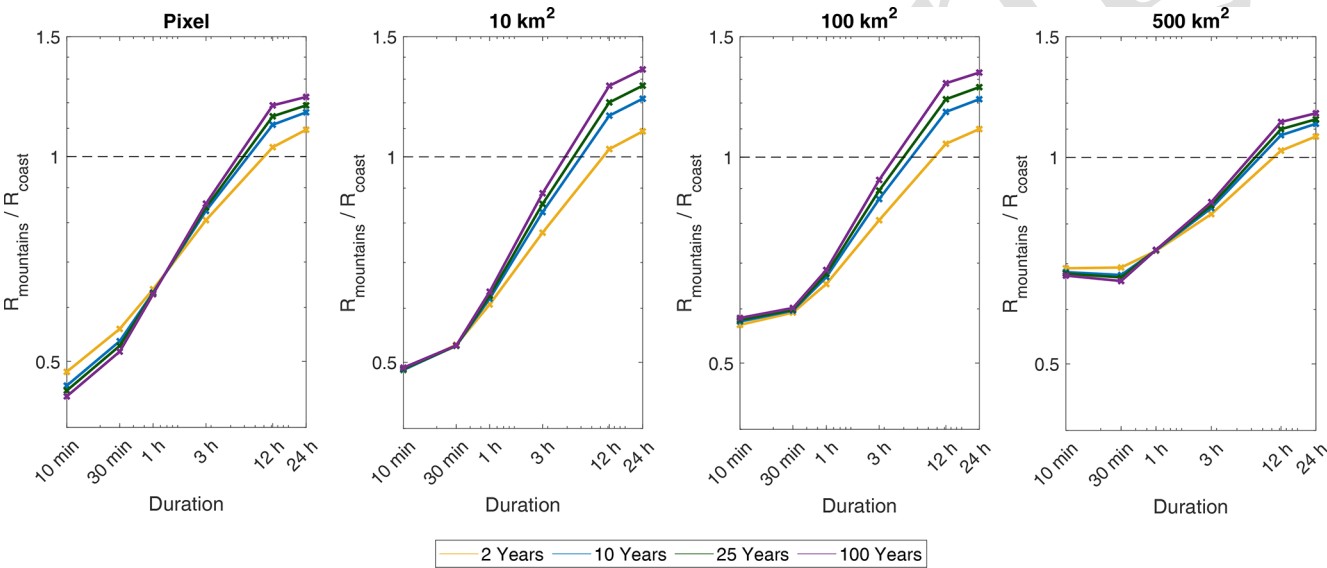

**Figure 7.** Ratio of return levels over the mountains compared to the coast for varying area sizes and return periods. The locations of the analysis sites are displayed in Fig. 1.

proximity to the coastline and orography on extreme precipitation.

## 5.1   Influence of area and duration on precipitation intensity similarity

Figure 5 presents the 25-year return levels derived for the coast, desert, and mountain locations (see Fig. 1), for varying area sizes. The return levels for 10- and 100-year return periods are presented in Fig. S2. The figure demonstrates that, at short durations, precipitation intensity is dissimilar over different area sizes, with precipitation intensity sharply decreasing with increasing area. However, as duration in-

creases, the return levels for different areas become indistinguishable, given the estimation uncertainty. Indeed, examining 24 h duration precipitation, intensity from the pixel scale up to 500 km$^2$ is almost identical for all area sizes. This observation holds true for all three locations considered, as well as for all the calculated return periods, despite the differences in geography and precipitation characteristics. Additionally, return levels were examined for the six validation sites (Fig. 1) to verify that this pattern is consistent throughout the study region, and the same pattern was consistently observed.

This convergence is attributed to the varying precipitation characteristics for short- and long-duration precipitation at

different scales. For short durations and small areas, rainfall is more localised; a single rain cell can cover the entire averaged area, which is smaller than the spatial scale of the cell. Therefore, the precipitation has a high level of homogeneity over the area. As area size increases, the examined area exceeds the spatial scale of the storm, thus decreasing the degree of homogeneity and resulting in less similarity in the precipitation intensities. With increasing durations, the time window of precipitation accumulation likely contains multiple rain cells, resulting in increasingly similar precipitation intensities for all area sizes.

The finding aligns with work presented by Peleg and Morin (2012), who examined the spatiotemporal characteristics of convective rain cells over northern Israel using a cell tracking algorithm. The study, based on high-resolution weather radar data, found that the mean area of the convective rain cells ranged between 36.6 and $64.4 \, \text{km}^2$, depending on the synoptic type (shallow low, Cyprus low, or active Red Sea trough), with a mean lifetime between 9.2 and 14 min and a maximum lifetime between 110 and 190 min. Belachsen et al. (2017) used the same cell-tracking algorithm to analyse precipitation from radar images over the Dead Sea area. They found that rain cells had a mean lifetime between 17.7 and 20.8 min depending on the synoptic type and an average cell area between 77.2 and $100.4 \, \text{km}^2$. Note that Belachsen et al. (2017) used a lower threshold of precipitation intensity for rain cell identification (a minimum of $5 \, \text{mm h}^{-1}$ vs $10 \, \text{mm h}^{-1}$ used by Peleg and Morin, 2012); thus their results are not directly comparable.

These studies confirm that the majority of rain cells in the study region have a lifetime under 30 min and an area size smaller than $100 \, \text{km}^2$, supporting the theory that the examined area will generally exceed the spatial size of the cell for areas $100 \, \text{km}^2$ and upwards and contain multiple rain cells for longer durations. Note that Peleg and Morin (2012) limited their study to northern Israel, which only covers the coast site analysed here, while Belachsen et al. (2017) only focused on the Dead Sea region. Additionally, both studies utilised a precipitation intensity threshold, whereas no lower limit was applied here, although low-intensity events will have been removed during the left-censoring of the data.

It is noteworthy that the estimated return levels for different spatial scales converge at different durations for the different regions (around 1 h over the desert and approximately 3 and 12 h over the coast and mountain regions, respectively). In desert areas, rainfall primarily stems from highly localised small-scale convective rain cells, and events are generally of short duration (Armon et al., 2020; Marra et al., 2017). Indeed for short durations, the highest rain intensity amounts in the region are located in the desert. Therefore, rainfall is very dissimilar at different spatial scales when considering short durations. At durations greater than 1 h, rainfall in desert areas becomes more homogenous in space, with less significant variations in rainfall intensity, causing this convergence. In contrast, rainfall events in the Mediterranean

coastal and mountainous regions generally have larger rainfall amounts for longer durations (Armon et al., 2020). The estimated return levels exhibit significant spatial differences for longer multi-hour durations and do not show homogenous behaviour over different spatial scales until around 3 to 12 h.

That rainfall becomes similar at long durations is significant as the transformation of point precipitation derived from rain gauge data to areal precipitation estimates is a topic of great interest. As discussed in Sect. 1, ARFs are generally used to estimate areal precipitation from point precipitation. According to the results presented in Fig. 5, applying ARFs at durations longer than 3–6 h may be unnecessary as precipitation at the pixel and areal scale is very similar for all the area sizes analysed here up to $500 \, \text{km}^2$. This is advantageous as ARFs proposed for a particular location can vary significantly, due to many factors such as differences in the methodology utilised and differences in the dataset used. Furthermore, ARF estimates often contain significant uncertainty.

The notion of increasing ARF values with increasing duration (indicating more similar values for point and areal precipitation) is widely accepted and is consistent with prior studies (and evidenced in all of the studies mentioned hereafter); however, the extent of similarity between point and areal precipitation remains unclear, with diverse findings in the literature. Pavlovic et al. (2016), for instance, produced ARF curves for 1 and 24 h durations, for 2- and 100-year return periods, using data from Oklahoma, central USA. In line with our analysis, their results showed that 24 h ARF values are significantly closer to 1 than 1 h values, with 24 h, 100-year, $500 \, \text{km}^2$ values of approximately 0.95 and 1 h values of approximately 0.75. Similarly, Overeem et al. (2010) calculated ARF values of 0.95, 0.84, and 0.7 for $100 \, \text{km}^2$ rainfall events with durations of 24 h, 1 h, and 15 min, respectively.

Conversely, various studies have found a more significant difference between point and areal precipitation. A study by Biondi et al. (2021), investigating the Calabria region in southern Italy using both a fixed and moving-centre approach, found that although ARF values increase with increasing duration, the estimated values for 24 h precipitation over large areas are low – indicating a large difference between the point- and large-scale areal precipitation. Specifically, they derived values of approximately 0.27 and 0.45 for 1 and 24 h duration rainfall over a $500 \, \text{km}^2$ area using a fixed-centre approach and values of 0.34 and 0.53 when applying a moving centre approach. They do note, however, that ARF values show a much sharper decrease for shorter durations due to the small areal extent of the short-duration events, while events with a long duration tend to be characterised by sustained rain rates over larger areas, as expected.

Likewise, Kim et al. (2019) derived ARF values for the Korean Peninsula of approximately 0.89 and 0.37 for 1 h duration precipitation over areas of 10 and $530 \, \text{km}^2$ respectively and values of 0.92 and 0.7 for 24 h precipitation over the same area sizes. These results again demonstrate that rainfall becomes more similar with increasing duration, but they still

indicate differences between the small- and large-scale areal precipitation. Lastly, Sivapalan and Blöschl (1998) analysed ARF values for a precipitation regime in Austria; they present their results in term of the scaled catchment area (A/$\lambda^2$), where $\lambda$ is the spatial correlation length of precipitation. They also found a large difference between point- and large-scale areal precipitation; analysing 24 h duration precipitation only, they show that for 10-year return period precipitation, ARFs decrease significantly with increasing catchment size, with an ARF value of approximately 0.95 for events with a scaled catchment area of 0.1 and a value of approximately 0.24 for a scaled catchment area of 100.

It should be noted that there are several factors which may influence the variability in these ARF values, including location, seasonality, rainfall type, and geographical characteristics, all of which have been demonstrated to effect ARF estimates (Kao et al., 2020). Moreover, the studies apply different methodologies for ARF calculation (moving centre vs fixed centre approach) and utilise different precipitation data sources (radar data vs rain gauge) and varying record lengths, all of which have demonstrated effects on ARF values. The specific precipitation characteristics observed in Fig. 5 are relevant only for the study area and for the analysed spatial and temporal scales. The specific durations at which precipitation intensities become similar may vary across different regions of the world, influenced by the characteristics of the storm regime. However, we believe that the general behaviour of intensities becoming increasingly similar with longer durations is expected to remain consistent. This understanding highlights the need for region-specific analyses when assessing the similarity between point and areal precipitation intensities.

The calculated shape and scale parameters, after correction factors have been applied, are presented in Fig. S4. The effect of both duration and area is clearly visible: the scale parameter decreases with increasing duration and increasing area, with the values converging at long durations – mirroring the behaviour of the return levels presented in Fig. 5. In contrast, the values of the shape parameter do not become more similar for long durations. The parameter displays non-monotonic behaviour, with generally minimal change for duration between 10 min and 1 h and a decrease for durations between 1 and 6 h (implying an increasing tail heaviness). Additionally, there is a significant difference between the pixel and areal scales: very low parameters, between 0.4 and 0.75 (indicating heavy tails), are observed for area sizes greater than the pixel scale, especially over the desert and mountains, while exponential tales (i.e. values close to 1) are observed for the pixel scale.

## 5.2 Scaling invariance of precipitation

Figure 6 is presented to analyse the scale invariance of precipitation over different areal scales. That precipitation intensity satisfies a simple-scaling relationship was first observed by Sherman (1905) and has been shown by numerous studies since (Gupta and Waymire, 1990; Innocenti et al., 2017). Simple scaling implies that precipitation intensity and duration are linked by a power-law relation (the logarithm of precipitation intensity and the logarithm of duration are linked by a linear relation). Simple scaling is widely used in extreme precipitation analysis. The main practical application of this property is the temporal downscaling of extreme precipitation data, for example, the estimation of sub-daily extremes from daily data (Yamoat et al., 2023) or of sub-hourly extremes from hourly data (Lee et al., 2022). IDF relations can then be derived using the downscaled data (Nguyen et al., 2002).

The duration-scaling characteristics of precipitation have been well studied; however, research has generally focused on point precipitation from rain gauge data. Here we can examine how precipitation changes with duration over different areal scales. Figure 6 illustrates that, at the pixel scale, precipitation displays a scale invariance that is well approximated by simple scaling. However, as area size increases, this power-law relation weakens, with return levels decreasing less sharply with duration, particularly in mountainous and coastal regions. Here, we quantify this deviation from the simple scaling by the coefficient of determination, $R^2$, of the linear regression between the log-transformed intensity and duration, shown in Fig. 6e. The $R^2$ value decreases with area for all three locations, indicating a decrease in the linearity of the relation. Very similar behaviour is observed for the different regions, especially between the coast and mountains. Results here are displayed for 25-year return levels, with 10- and 100-year return periods presented in Fig. S3. The same behaviour (with very similar $R^2$ values) is observed for all return periods.

The change in scaling is theorised to be related to the properties of precipitation in the region; at the pixel scale over a time window of precipitation accumulation, it is likely that only a single rain cell is present. However, over larger areas, as the duration window increases, multiple rain cells may be present, resulting in this non-linear relationship. In desert regions, precipitation is convective, and the lifetime of precipitation cells is generally shorter; thus precipitation over longer durations decreases more significantly than in coastal and mountainous regions, as evidenced in Fig. 6.

For point- and small-scale ($10\,km^2$) precipitation, simple scaling can be used for the downscaling of low-resolution daily precipitation data to higher-resolution sub-daily data and for the subsequent derivation of IDAF relations. However, for precipitation over larger areas a simple scaling approximation becomes less suitable, and more complex methods are required for an accurate downscaling of the data.

## 5.3 Orographic effect

The presence of mountains is known to cause highly variable precipitation patterns (Barros and Kuligowski, 1998; Houze

Jr. et al., 2001; Haiden et al., 1992). Orography causes a lifting of air masses along the windward slope of mountains, enhancing water vapour condensation and cloud formation and so increasing the overall precipitation yield. Conversely, on the leeward side of the slope, precipitation is decreased, as moisture in the descending air has been reduced. This phenomenon of "orographic enhancement" has been well documented in many regions worldwide on long-duration (daily and multi-daily) precipitation amounts and extremes (Johnson and Hanson, 1995; Napoli et al., 2019; Roe, 2005). Recent studies have demonstrated that the effect is reversed for short-duration (sub-hourly and hourly) extreme precipitation, with precipitation intensity decreasing with increasing elevation (Allamano et al., 2009; Avanzi et al., 2015; Formetta et al., 2022; Marra et al., 2021; Mazzoglio et al., 2022)

Performing an analysis focusing on Mediterranean cyclones in the eastern Mediterranean, Marra et al. (2021) proposed that the reverse orographic effect occurs as short-duration events typically consist of individual convective cores. The presence of orography redistributes precipitation to surrounding areas and smooths the event structure, thus causing decreased extreme precipitation. In contrast multi-hour and daily events include sequences of convective and stratiform-like elements, which aggregate due to orography and so cause an overall increase in precipitation. This interpretation was further supported by Dallan et al. (2023) by an analysis of convection-permitting model simulations in the Alps. The IDAF relations presented here also confirm the presence of the reverse orographic effect; Fig. 6 demonstrates that return levels are lower in the mountain region compared the coast for shorter durations (10 min–3 h) but are higher for long durations (12 and 24 h).

Here we examine the effects of area and return period on the reverse orographic effect. The ratio of return levels over the mountains vs the coast is shown in Fig. 7 for various return periods from 2 to 100 years. Longer return periods amplify the magnitude of the reverse orographic effect, causing decreased short-duration and increased long-duration precipitation in comparison to the coast. Conversely, larger areas appear to reduce the impact of the reverse orographic effect, significantly increasing the ratio of precipitation over mountains compared to the coast at short durations (0.45 for 10 min 25-year return levels at the pixel scale and 0.68 for $500\,\mathrm{km}^2$) and slightly decreasing this ratio for long durations (1.18 for 24 h 25-year return levels at the pixel scale and 1.13 for $500\,\mathrm{km}^2$). This phenomenon is attributed to the characteristics of convective precipitation in the region. The presence of orography directly influences rain cells over smaller areas. As the size of the area expands, the ellipses contain multiple rain cells, increasing the heterogeneity of the precipitation and producing an averaging effect that somewhat mitigates the impact of mountains, thus leading to a decrease in the reverse orographic effect.

The coast was selected as a comparison site to ensure that the storm ellipses did not extend into the mountain region, thereby avoiding any potential influence of orography on the derived return levels. However, it is important to acknowledge that the coast has been demonstrated to also enhance precipitation in the study region (as discussed in Sect. 5.4). Furthermore, it should be noted that for large area sizes over the mountains, the storm ellipses may extend beyond the mountain range and into lowland areas, depending on the orientation. This aspect could impact the results but is inherent to the analysis of spatial precipitation.

## 5.4 Coastal and orographic effects on multi-scale return levels

The effect of coastal proximity and orography on precipitation are well documented. Research on precipitation and proximity to the coastline has consistently found higher levels of precipitation near the coast (Daniels et al., 2014). The effect of elevation on precipitation, due to orographic forcing, has also been well studied (Guan et al., 2005; Lassegues, 2018; Tang et al., 2018; Marra et al., 2022). Marra et al. (2022) demonstrated that orography and the distance from the coastline influence extreme precipitation statistics and design precipitation intensities over the studied region. Analysing radar data on the pixel scale, they showed that at short durations (sub-hourly), return levels peak within a $\sim 20\text{–}40\,\mathrm{km}$ strip around the coastline and over the rift valley east of the mountain region. For longer durations, this peak in return levels moves further inland, corresponding to orographic ascents, and the rift valley causes decreased values.

Longitudinal variations in the return levels are examined here to see the effect of changing area on these coastal and orographic effects. The rationale for examining longitudinal transects relates to the typical advection direction of Mediterranean cyclones, which represent the vast majority of the storms, across the region. Two longitudinal transects (Fig. 1) are analysed, characterised from west to east by a sea–land boundary, a mountain range, a major valley, and a second mountain range. The location of these transects was selected by Marra et al. (2022) based on radar visibility and on the presence of regular orographic profiles. The transects are obtained by averaging the values over a 10 km region surrounding the latitudes.

The results are shown in Fig. 8, with longitudinal variations in the precipitation intensity distribution parameters along the same transects presented in Fig. S5. For 1 h durations, return levels peak around the coastline; this is generally most significant for smaller areas. This peak corresponds with an increase in the scale parameter for areas up to $100\,\mathrm{km}^2$. As found by Marra et al. (2022), the peak in return levels moves east for 24 h duration precipitation, reaching the first orographic barrier. This occurs for all area sizes, with a corresponding peak in both the scale and shape parameters (see Fig. S5). Marra et al. (2022) found that return levels also peak around the first mountain barrier. This peak

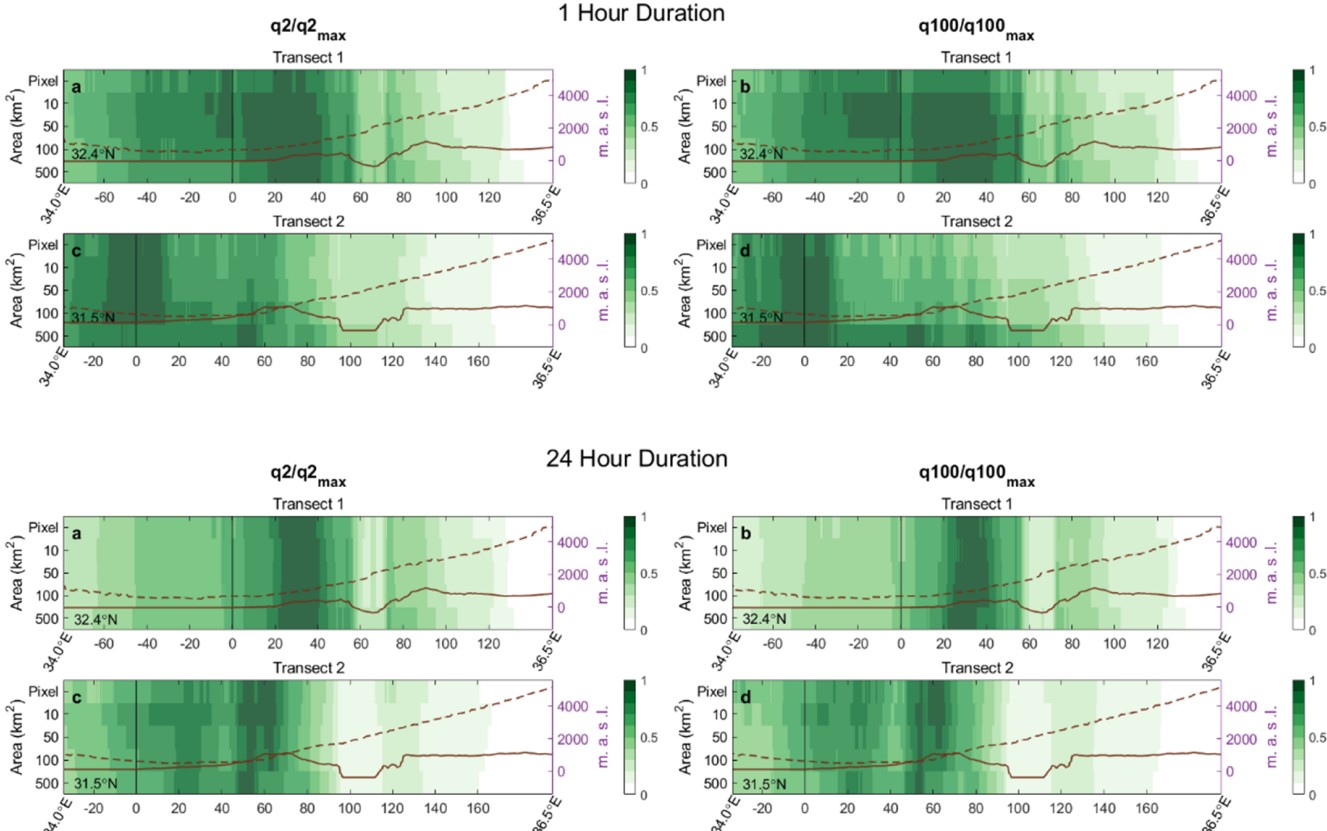

**Figure 8.** Longitudinal variations in 2- and 100-year precipitation return levels along the two transects shown in Fig. 1 as a function of coastline proximity (*x* axis) and area size (*y* axis). Results are shown for 1 h and 24 h durations. The transects are obtained by computing an average over the 10 km region surrounding the two latitudes. Return levels are then normalised over the maximum value along the transect to produce comparable values. Solid lines represent the orographic profile (see right-hand *y* axis). Dashed lines represent the sampling height of the lowest non-blocked radar beam (see right-hand *y* axis).

becomes wider (i.e. covers a greater distance) with increasing area. However, this widening can be explained by the way we explore large areas using ellipses: larger ellipses are likely to include orographically enhanced precipitation over the mountain areas from further distances away. For transect 1 there is also a peak at the second orographic barrier, which again widens with increasing area.

The rift valley results in decreased return levels for all area sizes, though this effect is more marked for 24 h durations than 1 h. The decrease in return levels appears smaller for the larger 100 and 500 km$^2$ areas. For transect 1, this corresponds to increased scale and shape parameters and for transect 2 a decreased shape parameter only. For sub-hourly durations, particularly 10 min (not shown here), the rift valley causes increasing values, as shown by Marra et al. (2022). This effect is more significant for larger areas.

The return level transects are shown for 2-year (a and c) and 100-year (b and d) return periods. With an increasing return period, there is generally similar behaviour, although the peak in return levels over the mountains is more prominent for 100 years.

## 6  Conclusions

The yearly exceedance probability of extreme precipitation at different spatial and temporal ranges is crucial for infrastructure design, risk management, and policymaking. This study applied the novel SMEV framework to estimate extreme precipitation return levels for multiple areas (0.25–500 km$^2$) and durations (10 min–24 h) directly from gauge-adjusted weather radar precipitation estimates. We focus on a region with sharp climatic gradients, characterised by a wide variety of climatic conditions. The application of the SMEV approach reduces uncertainties and enables the use of 12 years of radar record, obtaining estimates in line with those derived from averaging information from 30-year recording stations. Intensity–duration–area–frequency (IDAF) relations were derived from the estimated return levels and used to examine the climatological differences in precipitation intensity emerging from coastal, mountainous, and desert regions at different spatial and temporal scales. Three key points were discussed:

i. Precipitation and duration exhibit simple scaling at the pixel scale, but this relationship breaks down with increasing area – this has significance for temporal downscaling.

ii. Precipitation intensity is dissimilar for different area sizes at short durations but becomes increasingly similar at long durations – thus areal reduction factors may be unnecessary when computing precipitation for long durations.

iii. The reverse orographic effect is demonstrated to cause decreased precipitation for hourly and sub-hourly durations; however, this effect decreases over larger areas.

Overall, the study demonstrates that radar-precipitation-derived extreme return levels can provide important information for the understanding of extreme precipitation climatology at multiple temporal and spatial scales. Further, this information can be used for hydrological research and practice, as it provides two important innovations compared to the standard analysis from gauge station data: firstly, radar data allow the incorporation of the areal component into the analysis of extreme precipitation, and secondly it enables the derivation of IDAF spatial patterns at high resolution over the analysed region. Lastly, the study highlights the effectiveness of radar precipitation in deriving extreme return levels even in ungauged locations, broadening the application of extreme precipitation frequency analysis beyond the limitations of gauge station networks.

Our future research will focus on the identification of the spatial–temporal scales most relevant for extreme flood responses in catchments characterised by different sizes and climates. This will provide information useful toward a more practical application of these results in engineering and risk management.

*Code availability.* Codes used for the estimation of SMEV parameters and return levels are freely available at https://doi.org/10.5281/zenodo.3971558 (Marra, 2020).

*Data availability.* Rain gauge data were provided and pre-processed by the Israel Meteorological Service and are freely available at https://ims.gov.il/en/data_gov (last access: 10 April 2024, Israel Meteorological Service, 2024a). Original weather radar data were provided by the Israel Meteorological Service (https://ims.gov.il/en/node/179, last access: 22 June 2023, Israel Meteorological Service, 2024b). Corrected and gauge-adjusted radar data are available upon request to the head of the Hydrometeorology lab at the Hebrew University of Jerusalem, Efrat Morin (efrat.morin@mail.huji.ac.il).

*Supplement.* The supplement related to this article is available online at: https://doi.org/10.5194/hess-28-1-2024-supplement.

*Author contributions.* FM, EM, and TR conceptualised the paper, and TR conducted the analysis. TR prepared the manuscript with contributions from all co-authors.

*Competing interests.* At least one of the (co-)authors is a member of the editorial board of *Hydrology and Earth System Sciences*. The peer-review process was guided by an independent editor, and the authors also have no other competing interests to declare.

ther geographical representation in this paper. While Copernicus Publications makes every effort to include appropriate place names, the final responsibility lies with the authors.

*Acknowledgements.* Many of the methods used to create the weather radar archive were developed by the team in the past years, in addition to the novel statistical methodology used for extreme value analyses. Similarly, studies about extreme precipitation in the region and about the use of remotely sensed datasets for precipitation frequency analyses were published by the team. The number of self-citations is thus large. We did our best not to inflate the number of self-citations, and we hope the text will clarify why these citations are needed.

*Financial support.* This research has been supported by the Israel Science Foundation (grant nos. 1069/18 and 1999/22) and by the Center for Interdisciplinary Data Science Research at the Hebrew University of Jerusalem (CIDR). This study is a contribution to the HyMeX programme. Francesco Marra was partially supported by the CARIPARO Foundation through the Excellence Grant 2021 to the "Resilience" Project and by the COST Action CA19109 "Med-Cyclones" supported by COST European Cooperation in Science and Technology.

*Review statement.* This paper was edited by Marie-Claire ten Veldhuis and reviewed by two anonymous referees.

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

**Remarks from the language copy-editor**

CE1 Please provide an explanation of why this needs to be changed. We have to ask the handling editor for approval. Thanks.

CE2 Please provide an explanation of why this needs to be changed. We have to ask the handling editor for approval. Thanks.

CE3 Please confirm that all figures in the paper are correct prior to publication.