# Peer review of "Exploring patterns in precipitation intensity-duration-area-frequency relationships using weather radar data"

_EGUsphere, 2023_

## Author Comment (AC1)

**Response to review 2**

We extend our sincere gratitude for the dedicated time and effort you invested in reviewing our manuscript, are we are grateful for the constructive comments on and valuable improvements to our paper. We have incorporated changes to reflect the suggestions provided. Please see below, in blue, for a point-by-point response to the comments and concerns.

The paper explores precipitation patterns and rainfall statistics in space and time based on weather radar data and the application of the method SMEV. This method is described in detail in Marra et al (2022) for radar pixel values however here the method is extrapolated to also include spatial rainfall and thus dependence of area.

The paper is generally well-written and understandable. Several things are indeed interesting – especially the pixel-area relations depending on rainfall duration as well as the significant climatological differences in the study area.

Thank you very much.

1) The novelty of the paper should be emphasized more in the introduction. I guess compared to former efforts, the novelty here is the application of SMEV also on the areal component rather than single pixels?

Author response: Thank you for the suggestion. Yes, this is the main novelty. The follow has been added to the paper to emphasise this:

*"In this study we apply the SMEV framework to examine extreme precipitation at various spatial scales for the first time, in order to investigate the impact of area size on local extremes. As stated, while the SMEV framework has demonstrated efficacy in successfully estimating extreme rainfall, its prior applications have been confined to either the point (in the case of rain gauge data) or pixel (when utilising radar rainfall data) scale analyses at different temporal scales. We here extend the application of the SMEV to estimate extreme return levels up to 100 years across multiple spatial and temporal scales."*

2) In my view, it is not clear what the differences between MEV and SMEV are. The authors refer to past publications, but could be relevant to describe the SMEV in a bit more detail here in order to understand how it differs from other methods of extreme value statistics. For example line 157-158 could be detailed further.

Author response: Thank you for the suggestion. The text has been revised and now reads:

*"Extreme precipitation return levels are estimated across the study area using the novel non-asymptotic SMEV framework proposed by Marra et al. (2019a) and (2020), a simplified version of the original MEV framework proposed by Marani*

*and Ignaccolo (2015). The MEV and SMEV approaches are based on the concept of 'ordinary events', which are all the independent realisations of the process of interest. Unlike classic extreme value theory, which only exploit a small subset of the data, i.e. the annual maxima or the peaks exceeding a high threshold, they make use of a greater proportion of observations to fit the distribution parameters, thus decreasing the parameter estimation uncertainty. The SMEV is a modified version of the MEV; it neglects the interannual variability of the distribution of ordinary events and in their number of yearly occurrences (Marra et al., 2019a). The SMEV formulation significantly reduces the number of parameters and allows for a direct interpretation of their meaning. This results in a simpler formulation for the non-exceedance probabilities of extreme rainfall, and more robust parameter estimation. Several studies have applied the SMEV to precipitation frequency analysis over different regions (Marra et al., 2020, 2019a; Miniussi and Marra, 2021; Araujo et al., 2023), including over the study area (Marra et al., 2022), and have demonstrated the robustness of the method's assumptions and its ability to reproduce extreme frequencies from relatively short records. The SMEV is used here to estimate precipitation events of varying sizes and durations, so that spatial and temporal effects on extreme precipitation can be analysed"*

3) The equation in line 128: Shouldn't it be Z=316R^1.5?

Author response: Yes, thank you, this has been corrected

4) Do you think the discrepancy between gauge and radar is due to differences in climatology, bias adjustment, or radar artifacts? Consulting figure 4 it seems that there are some radar issues close to the radar – maybe something related to scanning and CAPPI generation?

Author response: We cannot be sure of the exact reason for the discrepancy, however we hypothesise that it is due mainly to issues with the radar. There are no substantial differences in climatology in this area, and there are no indications in the correction factors that they are causing the increased values here. There is, however, no CAPPI generation via interpolation or similar in this radar archive, so CAPPI generation can be excluded from the causes. As stated in the text:

*"Despite best efforts, the exact issue with the data here could not be fully identified by the authors. It is suggested that the poor results may be due to the close proximity of the site to the radar station, where radar precipitation intensity estimates are known to be poor and unreliable, or to other types of radar systematic errors (e.g., residual ground echoes)."*

The following was also added to the text:

*"There are no substantial changes in climatology in this area that could cause these discrepancies. Additionally, there are no indications that the parameter correction factors are related to the increased values here."*

The area of low values around the radar station is indeed caused by the close proximity to the radar station and this is stated in the text:

*"The area of very low values, attributed to data quality issues around the radar location, is clearly visible."*

5) Figure 5. I think it would be very interesting also to include the rain gauges statistics in this figure – and in principle also in figures 6-7. Typically, you would see an underestimation of the radar estimates at short durations. See e.g. Schleiss et al (2020) or Andersen et al 2021. The point scale is indeed interesting to study in addition to the study of the difference between pixel scale and areas of 10, 100 and 500 km2. In regards to point 1 in the conclusion the actual comparison in the manuscript is not made on point scale but on a pixel scale. Even going from point to pixel scale will result in scaling – which will be more dominant for shorter durations than larger ones.

Author response: Thank you for this comment. Actually, radar return levels are adjusted to exactly match the rain gauge ones at the pixel scale (see Marra et al., 2022). This means that the effects of areal mismatch between rain gauges and radar cannot be seen in the estimated return levels. The referee is kindly referred to Marra et al. 2022 for an analysis and discussion of the mismatch between weather radar at the pixel scale and rain gauges (see Fig. 2 in the mentioned paper and related discussion). As this was already examined in the previous study, we keep it out of the objectives of this study.

Thank you for raising the terminology mismatch, the "point" has been corrected to "pixel" throughout the text. Although we note that, as mentioned above, after adjustment the pixel scale is representative of the rain gauge (i.e., point) scale.

6) It would be relevant to present fitted parameter values of shape, scale, kappa, lambda in order to study how the vary over different durations and area. Maybe present selected values in a table. Potentially an empirical relation between model parameter estimates and duration and area could be sought.

Author response: Thank you for the suggestion, which is in line also with Reviewer 1 comment. We have therefore added a figure (rather than table) showing the shape and scale parameters (see figure below) along with the following text:

*"The calculated shape and scale parameters, after correction factors have been applied, are presented in Fig. S4. The effect of both duration and area is clearly visible: the scale parameter decreases with increasing duration and increasing area, with the values converging at long durations – mirroring the behaviour of the return levels presented in Fig. 5. Unlike the scale parameter, the values of the shape parameter do not become more similar for long durations. The parameter displays non-monotonic behaviour, with generally minimal change for durations between 10 min and 1 h, and decreasing for durations between 1 and 6 h (implying an increasing tail heaviness). Very low parameters, between 0.4 and 0.75 (indicating heavy tails), are observed for*

*area sizes greater than the pixel scale, especially over the desert and mountains, while exponential tales (i.e. values close to 1) are observed for the pixel scale."*

[Figure]

*Shape and scale parameters (after correction factors have been applied) as a function of area and duration estimated for the desert, coast, and mountains. Shaded areas represent the 90 % confidence interval from 100 bootstrap samples.*

---

## Author Response (AR1)

**Response to review 1**

We wish to express our appreciation for the time and effort you have dedicated to providing feedback on our manuscript. The in-depth comments, suggestions, and corrections have been immensely helpful and have greatly improved the manuscript. We have incorporated changes to reflect the suggestions provided. Please see below, in blue, for a point-by-point response to the comments and concerns. All page numbers refer to the revised manuscript file without and with tracked changes.

The authors explore the patterns of rainfall intensity-duration-area-frequency (IDAF) curves derived from adjusted weather data on the eastern Mediterranean region. IDAF curves at durations from 10min to 1 day and area from $0.25km^2$ to $500km^2$ are derived by applying the simplified meta-statistical extreme value analysis (SMEV) on 18 years of available weather radar data. Their study concludes that the simple scaling of rainfall intensities with duration is only valid for point scale, that area reduction factors are mainly useful for short durations (<3h), and that the reverse orographic effect is weaken with larger areas. Overall, I find the study relevant, very well written and easy to read/follow.

Thank you very much

However, I have still some recommendations or points that I would like to discuss with the authors regarding the study:

1. Figure 1 is a bit difficult to understand because it contains so much information. I would suggest that the background colour shows the land elevation and that the climate classification is given in semi-transparent polygons or lines. The size of the rain gauge-points can be a bit bigger so we can distinguish them better. Maybe the x and y axis for the right part of the figure can show the distance in meters from the weather radar location.

Author response: Thank you for the suggestions, we agree that the figure is busy and hard to understand. The size of the rain gauge markers has been increased and the background changed to show land elevation, with the climate classifications presented as lines legend (see figure below).

We prefer to keep the x- and y-axis as latitude and longitude as we feel that this is important information for the reader. However, we have moved the scale marker so that it aligns with the radar location, so that the distance from the radar can be better intuited. Additionally, we added the distance in km of the radar extent (140 km) to the figure's legend. We considered adding distance circles at 50 and 100 km from the radar, however we found this to be too busy.

[Figure]

*Map of the study area*

2.  I'm a little bit confused with the correction and adjustment of the radar data. So as far as I understood there are in total 3 adjustments performed to radar data based on the rain gauge information. So with the two first adjustments you are trying to adjust rainfall intensities (using daily stations), and then with the third one you are adjusting directly the SMEV parameters (using the 10min stations). I am wondering if all three steps are necessary and not redundant, since in the end you adjust the SMEV radar parameters according to the 10min station parameters. Could you please comment a bit more on the necessity of these three adjustments? Do you know how much the SMEV radar parameters are changing due to the correction based on daily stations (i.e. if you do only adjustment 3 vs adjustment 2 and 3, vs all adjustments together?

Author response: Thank you for pointing this out. Indeed, there are three steps in the adjustment, but they are not redundant. The first two steps are done during the creation of the radar database. They optimise the radar archive in 'average' terms meaning that they aim at providing an archive that is as good as possible for (i) any moment and (ii) any location in the domain in terms of bias and dispersion of the daily precipitation amounts (see details in Marra and Morin, 2015 and Marra et al, 2022). These adjustments are not part of this study, they come with the radar archive and cannot be undone.

To make this aspect clearer, we rephrased the text of Section 2.1 to:

*"The final radar archive was obtained after a two-step bias adjustment based daily rain gauge archive data (Morin and Gabella, 2007; Marra and Morin, 2015). This*

*adjustment aimed at optimising the bias and dispersion of rainfall depths during independent meteorological events. A full description of the radar data elaboration procedure and overall quality is provided in Marra et al. (2022). Marra et al. (2022) demonstrated…".* [Line 137/148]

Clearly, an adjustment optimised for the average conditions will not necessarily be optimised also for extremes. The last adjustment we apply follows the approach by Marra et al. (2022) and aims at optimising the estimation of return levels from the radar based on ground "truth" from high-resolution rain gauge records. This adjustment operates in the SMEV parameter space and only adjusts the SMEV parameters and thus estimated return levels, not the weather radar estimates or the return levels estimated with any other extreme value method. We believe the text edit above clarifies this issue.

3. Also regarding the parameter scaling of different areas based on rainfall stations, do you know how drastic the change in the parameters of the bigger areas is? It would be interesting to see how the mean value of the parameters of different duration and area are changing after the adjustment. So to have an idea how "wrong" the radar parameters are, and which duration and areas are mostly affected by it. Maybe this could explain also the convergence of the IDAF curves for longer durations? On the other hand, it would be also interesting to see what parameters are mainly differing with station based parameters (either shape, or scale or the number of ordinary events). This is probably outside the scope of your study, but maybe you can give your insight in the discussion based on your experience so far.

Author response: The parameter scaling was developed originally for the pixel scale and has here been applied 'as is' to the areal scale – therefore the scaling is the same for each area size, and differs only for duration. Developing a scaling method for the areal scale would be preferable, but is not possible here due to the low density of rain gauges.

An analysis of the parameter correction factors shows that the mean of the absolute scaling factors over the study area for both the shape and scale parameters are slightly greater for smaller durations than larger durations, ranging from 1.33 (24 h) to 1.46 (10 min) for the mean scale correction factor and 1.23 (24 h) to 1.58 (10 min) for the mean shape correction factor. However, these differences are relatively minor. Additionally, the scaling is similar for the shape and the scale parameters. The correction factors for the number of ordinary events have a mean value of 1.75 (note that the correction factors for the number of ordinary events do not change with duration). The spatial distribution of the correction factors for the shape, scale, and n parameters varies across the study area. However, the patterns of variation are generally consistent for each duration.

Based on this it seems unlikely that the scaling of the parameters is causing the convergence observed at long durations.

4. Another thing that is not completely clear to me, is the identification of storms and ordinary events at the pixel scale. So you first determine the storm events based on daily average data (a total of 498 storm events). Then at each pixel for these storm events are you; a) either defining new "local" storm events, that can have completely different durations than the "regional" ones, or b) are you just checking which "regional" storms are manifested in this pixel and then decide whether to exclude them or not (but you keep the event duration same). I am asking because the events for each pixel are based on 10 min radar data, and it may be that the duration of such events is shorter than 24 hours (which would them compromise your fixed number of events over different durations).

Author response: The events are identified using method b. Indeed, it is likely that events are shorter than 24 hours, but this is not a problem with respect to the fixed number of events across durations. This is thanks to the unified approach we use to define the ordinary events, which goes through the identification of independent "storms" separated by at least 24 dry hours. This separation grants that, using a moving window of 24 hours (i.e. examining 24-hour durations), storms lasting for short (e.g., only for 1 hour in the pixel of interest) would still yield independent ordinary events for all durations up to 24 hours. Note that for storms lasting less than 24 h, when considering 24 h duration ordinary events the time window containing the maximal intensity will therefore include zero values. This holds true for all other durations as well.

Moreover, the annual maxima computed over moving windows of the desired duration would be a subset of the corresponding ordinary events. This approach is recently being widely adopted for multi-duration precipitation frequency analyses (e.g., Marra et al., 2020; Marra et al., 2021; Dallan et al, 2022; Marra et al., 2022; Formetta et al., 2022; Dallan et al., 2023; Araujo et al., 2023; Shmilovitz et al., 2023); we kindly refer to Marra et al. (2020), where it was introduced, for further details.

Throughout the paper the text was amended to emphasise the difference between storms and ordinary events. Additionally, the following text was added to clarify that the number of events is the same for all durations:

> "Ordinary events at the spatial (area) and temporal (duration) scales of interest are then identified at each radar pixel for each storm, with one ordinary event calculated for each storm… It should be noted that for each area considered, the number of ordinary events at each pixel is consistent for all the examined durations. This is due to the unified approach used to define the ordinary events, which goes through the identification of independent 'storms' separated by at least 24 dry hours." [line 181/196]

Araujo D, F Marra, H Ali, HJ Fowler, EI Nikolopoulos, 2023. Relation Between Storm Characteristics and Extreme Precipitation Statistics Over CONUS. Adv. Water Resour., 178, 104497, https://doi.org/10.1016/j.advwatres.2023.104497

Dallan E, F Marra, G Fosser, M Marani, G Formetta, C Shäer, M Borga, 2023. How well does a convection-permitting regional climate model represent the reverse orographic effect of extreme precipitation? Hydrol. Earth Sys. Sci., 27, 1133-1149, https://doi.org/10.5194/hess-27-1133-2023

Dallan E, M Borga, M Zaramella, F Marra, 2022. Enhanced summer convection explains observed trends in extreme subdaily precipitation in the Eastern Italian Alps. Geophys. Res. Lett., 49, e2021GL096727. https://doi.org/10.1029/2021GL096727

Formetta G, F Marra, E Dallan, M Zaramella, M Borga, 2022. Differential orographic impact on sub-hourly, hourly, and daily extreme precipitation. Adv. Water Resour., 149, 104085, https://doi.org/10.1016/j.advwatres.2021.104085

Marra F, M Armon, M Borga, E Morin, 2021. Orographic effect on extreme precipitation statistics peaks at hourly time scales. Geophys. Res. Lett., e2020GL091498, https://doi.org/10.1029/2020GL091498

Marra F, M Armon, E Morin, 2022. Coastal and orographic effects on extreme precipitation revealed by weather radar observations. Hydrol. Earth Syst. Sci., 26, 1439–1458, https://doi.org/10.5194/hess-26-1439-2022

Shmilovitz Y, F Marra, Y Enzel, E Morin, M Armon, A Matmon, A Mushkin, Y Levi, P Khain M Rossi, G Tucker, J Pederson, I Haviv, 2023. The impact of extreme rainstorms on escarpment morphology in arid areas: insights from the central Negev Desert. J. Geophys. Res.: Earth Surface, 128, e2023JF007093, https://doi.org/10.1029/2023JF007093

5. Following the explanation on line 210-211, is the number of ordinary events reduced according to the 55th percent, or just the input series for the CDF fitting is reduced to leave out the 55% of the events? Also, in Line 210 you mention than censoring between 55th to 80th quantile doesn't influence much the results, but then still why did you choose to censor below the 55th quantile?

Author response: From the referee's question we understand that the meaning of left censoring was not completely clear. As now better detailed in the text (see below) the left censoring procedure ignores the intensities of the censored events during the fitting of the CDF, but it is important to note that the weight of the censored events is retained in probability. The text has been revised to include this and reads as follows:

*"The left censoring procedure ignores the intensities of the censored events while still retaining their weight in the probability. The study found that left-censoring values between the 55th quantile and the 80th quantile provide virtually indistinguishable results for the area. Following Marra et al., (2022) we here left-censor the lowest 55 % of the ordinary events."* [line 222/238]

We selected the lower threshold (the 55th quantile) so as to include the maximum number of events in the data sample, to reduce uncertainty. The following was added to the text:

*"The lower threshold was selected to include the maximum number of events in the data sample."* [line 225/241]

6. Line 240, could you please describe shortly the bootstrapping from Overeem et al. 2008? Does it pool together all stations inside a region and samples from pooled storms, or is it just storm sampling with replacement from a single series?

Author response: The bootstrapping procedure performs sampling with replacement of years from a single series, and is applied independently to each pixel. The procedure is as follows:

The technique generates samples by selecting blocks (here blocks are defined as a hydrological year) randomly with replacement, so that the number of blocks is the same as in the original record. The ordinary events for each block are then concatenated to create the bootstrapped dataset, from which the Weibull parameters and quantiles are estimated, using the procedure described above. This enables the block structure of the original rainfall data to be preserved.

The above explanation has been added to the text [Line 255/271].

7. Lines 251-252, why do you validate the radar data based on station data of another time period? Wouldn't this also punish more the radar data IDAF curves?

Author response: This is correct, however, we wanted to thoroughly asses the accuracy of the radar derived return levels and therefore wanted to ensure that the comparison rain gauge derived results were as accurate as possible. We thus decided to use the entire dataset. The following was added to explain this in the text:

*"The rain gauge data spans a 30-year period, in contrast to the 12-year dataset used to derive the radar data results. It was decided to use the whole timeseries, rather than matching the time periods, so as to produce the most accurate return levels against which to validate the radar derived results."* [line 283/298]

It is also correct that this will bias the rain gauge data towards producing better results than the radar data. However, as the results show (figure 3), generally the uncertainty of the radar derived results are comparable to the gauge derived results, despite using less data. This is noted in the next:

*"This is encouraging as the radar results are computed using only 12 years of data and are adjusted using relations derived for the pixel-scale, whilst the gauge results utilise 30 years of data and direct precipitation observations."* [line 299/318]

8. At line 260, you mention the discrepancy between radar and daily station IDF curves due to different daily measurements. Since radar is at 10mins, wasn't it possible to calculate the daily maximum intensity according to the daily measurement times (between 6 am to 6 am)?

Author response: We thank the reviewer for the suggestion. We have calculated the return levels using 6am -6am daily radar data and have included these results in the paper. Interestingly, this 6am to 6am data produces no discrepancy between the radar and rain gauge results for sites e and f. We have added the following description and possible explanation for this:

> *"Distinctions arise between the 24 h and the 6 am to 6am daily radar derived results in certain regions. The 6am to 6am radar results generally show very similar behaviour to the rain gauge derived results for all 6 sites, as well as similar levels of uncertainty; indeed within the uncertainty intervals the radar estimates largely cannot be distinguished from the gauge estimates. For sites a, b, c, and d the 24 h radar data also produces good results, producing return levels very similar to the gauge derived levels. As expected, the 24 h return levels are higher than the 6am to 6am radar levels, as the exact time window maximising precipitation intensity for each storm is utilised, rather than the maximal 6am to 6am period. However, at locations e and f, the radar derived return levels significantly exceed the rain gauge derived levels. Interestingly, this mismatch is specific to these two locations, and the 6 am to 6 am radar data yields satisfactory results for these sites.*
>
> *An analysis by Marra et al. (2022) offers insight into this discrepancy, by examining the time of the day at which the highest short-duration intensities (i.e., the ordinary events in the distribution tail, as in this study defined as the largest 45 %) occur over the study area. They found that the highest offshore intensities tend to occur in the early morning (02:00–08:00 UTC) or morning (08:00–14:00 UTC), and then shift to mostly morning (08:00–14:00 UTC) at the coastline and near inland. This is caused by the convergence created by the superposition of the westerly winds typical of Mediterranean cyclones with land breeze, which is expected to peak in the early morning hours when the sea is the warmest compared to the land. Although Marra et al (2022) focus only on short duration rainfall, and the results are given here for 24 h events, these findings may still explain the discrepancy between the results. Given that sites e and f are situated on the coastline, high rainfall intensities occurring more often in the early mornings between 02:00–08:00 UTC, this could lead to large differences between the maximal 24-hour value and the maximal 6am to 6am event values. Site d is somewhat more east, with Marra et al. (2022) indicating peak rainfall between 06:00 and 08:00 UTC for this site, whilst sites b and c are the most inshore and present high rainfall intensity peak times of approximately 11:00 – 14:00 UTC and 08:00 – 11:00 UTC respectively. Therefore limiting the daily data to 6am to 6am may have a lesser impact on these inland sites."* [line 305/323]

9. Figure 3 – c, I agree it might be the distance to the radar station that is causing such overestimation. However, this pattern is not consistent with Figure 4, as we see that in the region near to the radar station there is a clear underestimation (or very little rainfall). I was wondering if there is a specific parameter that is overestimated in this area that might be directly link with this IDF overestimation?

Can it be that the adjustment to 10min station data parameters had something to do with the overestimation (like the density of 10min station data in the vicinity)?

Author response: The region around the radar station does have significant underestimation of the rainfall. However, the overestimation that can be seen in location e is the region of very high values just south of this area (visible in figure 4). This is stated in the text:

*"Immediately south of the radar station there is also a distinct region of high values. This corresponds to the location of validation site e shown in Fig. 1."* [line 335/371]

After calculating the return levels using 6am to 6am daily data it now appears that these results are not overestimated (as explained in comment 8).

10. At section 4.3 you explain how the figures are derived, however you mention in line 313 a 5 by 10 $km^2$ box, and then on line 319 a 10 by 10 $km^2$ Is this a typo, or these are actually two different types of box-sizes used for the investigation?

Author response: Thank you for pointing this out, it was a typo that has now been corrected. The correct size is 10 by 10 $km^2$.

11. Figure 5 – I think it is also interesting to point out the duration when the areas converge for these three regions. In the desert the convergence happens at 1 hour, while at coast and mountains at 3 hours. Do you have any explanation for that? Maybe to explain why the IDAF curves are converging after a certain duration, it may be useful to have a look at the SMEV parameters and see how they are changing with duration and area, or even see the average characteristics of the ellipses for each duration and area; so for instance if for 24 h duration the axis ratio of the ordinary events is closer to 1 than those of 1h duration, or even the spatial variability of the rainfall within an ellipse for different durations and areas.

Author response: Yes, this is an interesting observation. We think it is probably to do with the nature of the rainfall in the different regions. The following was added:

*"It is noteworthy that the estimated return levels for different spatial scales converge at different durations for the different regions (around 1 h over the desert and approximately 3 and 12 h over the coast and mountain regions, respectively). In desert areas rainfall primarily stems from highly localised small-scale convective rain cells, and events are generally short duration (Armon et al., 2020, Marra et al. 2017). Indeed for short durations, the highest rain intensity amounts in the region are located in the desert. Therefore, rainfall is very different at different spatial scales for short duration. At durations greater than 1 h rainfall becomes more homogenous in space, with less significant variations in rainfall intensity, causing*

*this convergence. In contrast, rainfall events in the Mediterranean coastal and mountain regions generally have larger rainfall amounts for longer durations (Armon et al., 2020). The estimated return levels exhibit significant spatial differences for longer multi-hour durations, and do not show homogenous behaviour over different spatial scales until around 3 to 12 h."* [line 403/440]

A figure of the shape and scale parameters as a function of area and duration has been added to the supplement of the paper (see figure below), along with the following text:

*"The calculated shape and scale parameters, after correction factors have been applied, are presented in Fig. S4. The effect of both duration and area is clearly visible: the scale parameter decreases with increasing duration and increasing area, with the values converging at long durations – mirroring the behaviour of the return levels presented in Fig. 5. Unlike the scale parameter, the values of the shape parameter do not become more similar for long durations. The parameter displays non-monotonic behaviour, with generally minimal change for durations between 10 min and 1 h, and decreasing for durations between 1 and 6 h (implying an increasing tail heaviness). Very low parameters, between 0.4 and 0.75 (indicating heavy tails), are observed for area sizes greater than the pixel scale, especially over the desert and mountains, while exponential tales (i.e. values close to 1) are observed for the pixel scale."* [line 454/499]

[Figure]

*Shape and scale parameters (after correction factors have been applied) as a function of area and duration estimated for the desert, coast, and mountains. Shaded areas represent the 90 % confidence interval from 100 bootstrap samples.*

12. Also the results from Figure 5 are a bit controversial, as I would expect that the ARF are dependent on duration and area (see for instance Overeem et al. 2010), and in my opinion these results should be discussed more. Line 375-397 – here you are discussing about other studies that have more or less contrary results to your investigation. The main reason for this contrast, you list the different study areas. However, might there be other factors like the methodology applied or the data used that might explain the difference in the results (i.e. use of ellipses instead of circles, use of SMEV instead of GEV and so on). Lastly is the same pattern as shown in Figure 5 also valid for other locations, i.e. the validation sites or other random sites?

Author response: Thank you for pointing this out. In general, all the studies on ARF show that ARF values increase with increasing duration, indicating more similar behaviour between the point and the areal scale for longer durations. This agrees with the results of

figure 5 - for 24 h durations rainfall intensity is very similar for all areal scales, while for 10 min rainfall there is a large spread of values.

The disagreement in the studies is the extent of this similarity - which is described by the value of the ARF. The studies mentioned show a range of results. The text was revised to make this clearer and now reads:

> *"The notion of increasing ARF values with increasing duration (indicating more similar values for point and areal precipitation) is widely accepted and is consistent with prior studies (and evidenced in all of the studies mentioned hereafter); however, the extent of similarity between point and areal precipitation remains unclear, with diverse findings in the literature. Pavlovic et al. (2016) for instance, produced ARF curves for 1 and 24 h durations, for 2- and 100-year return periods, using data from Oklahoma, central USA. In line with our analysis their results showed that 24 h ARF values are significantly closer to 1 than 1 h values, with 24 h, 100-year, 500 $km^2$ values of approximately 0.95, and 1 h values of approximately 0.75. Similarly, Overeem et al. (2010) calculated ARF values of 0.95, 0.84 and 0.7 for 100 $km^2$ rainfall events with durations of 24 h, 1 h and 15 min, respectively.*
>
> *Conversely, various studies have found a more significant difference between point and areal precipitation. A study by Biondi et al. (2021), investigating the Calabria region in southern Italy using both a fixed and moving-centre approach, found that although ARF values increase with increasing duration, the estimated values for 24 h precipitation over large areas are low – indicating a large difference between the point and large-scale areal precipitation. Specifically, they derived values of approximately 0.27 and 0.45 for 1 and 24 h duration rainfall over a 500 $km^2$ area using a fixed centre approach, and values of 0.34 and 0.53 when applying a moving centre approach. They do note, however, that ARF values show a much sharper decrease for shorter durations due to the small areal extent of the short-duration events, while events with a long duration tend to be characterised by sustained rain rates over larger areas, as expected.*
>
> *Likewise, Kim et al. (2019) derived ARF values for the Korean peninsula of approximately 0.89 and 0.37 for 1 h duration precipitation over areas of 10 $km^2$ and 530 $km^2$ respectively, and values of 0.92 and 0.7 for 24 h precipitation over the same area sizes. These results again demonstrate that rainfall becomes more similar with increasing duration, but still indicate differences between the small and large-scale areal precipitation. Lastly, Sivapalan and Blöschl (1998) analysed ARF values for a precipitation regime in Austria, they present their results in term of the scaled catchment area (A/λ2) where λ is the spatial correlation length of precipitation. They also found a large difference between point and large-scale areal precipitation; analysing 24 h duration precipitation only, they show that for 10-year return period precipitation ARFs decrease significantly with increasing catchment size, with an ARF value of approximately 0.95 for events with a scaled catchment area of 0.1 and a value of approximately 0.24 for a scaled catchment area of 100."* [Line 419/456]

It is very likely that the different methodologies used to derive the ARF values, as well as a number of other factors, will affect the results – this has been demonstrated in a number of papers. The following was added to the paper:

> *"It should be noted that there are several factors which may influence the variability in these ARF values. The studies are focused on different locations, characterised by varying seasonality, rainfall types and geographical characteristics, all of which have been demonstrated to affect ARF estimates (Kao et al., 2020). Moreover, the studies apply different methodologies for ARF calculation (moving centre vs fixed centre approach), different precipitation data sources (radar data vs rain gauge) and varying record lengths, all of which have demonstrated effects on ARF values." [Line 444/483]*

To investigate if the same pattern seen in figure 5 is present for other locations the IDAF curves were plotted for the 6 validation sites (see figure below). It is evident that the same pattern exists. The following was added to the paper (without the figure):

> *"Additionally, return levels were examined for the six validation sites to verify that this pattern is consistent throughout the study region, and the same pattern was observed."* [Line 379/416]

[Figure]

*25 year return period IDAF curves estimated for the six validation sites.*

13. In Section 5.2 (more specifically starting from Line 410 and on) you mention that the power-law relation weakens as the area size increases. Do you know of any other study that might back you up in this conclusion?

Author response: The power law relation between intensity and duration is well covered in the literature. However, as far as we can tell the influence of area on the intensity duration relationship has not been previously studied. Similarly, there are also studies on the power law between intensity and area, related to the spatial scaling of rainfall, however, they do not consider the impact of duration.

A number of papers look at the effect of area and duration on areal reduction factors – for example Kim et al. (2019) show that ARF generally increases with duration, and that this phenomenon becomes pronounced as the area becomes larger, however they do not relate it to the power-law relationship.

14. It seems that this works is largely based on the previous work of Marra et al. 2022. Maybe you can consider to join this paper with the previous one, so readers will go directly to the previous one if they have any questions.

Author response: Thank you for the suggestion. Part of the methods are indeed taken from the work by Marra et al, 2022, but the scientific objective here is different as we focus on the areal dimension. For this reason, and since the above mentioned paper was published over 1 year ago by a slightly different team, we believe it is not appropriate to join the two papers (even if at all possible).

**Response to review 2**

We extend our sincere gratitude for the dedicated time and effort you invested in reviewing our manuscript, are we are grateful for the constructive comments on and valuable improvements to our paper. We have incorporated changes to reflect the suggestions provided. Please see below, in blue, for a point-by-point response to the comments and concerns. All page numbers refer to the revised manuscript file without and with tracked changes.

The paper explores precipitation patterns and rainfall statistics in space and time based on weather radar data and the application of the method SMEV. This method is described in detail in Marra et al (2022) for radar pixel values however here the method is extrapolated to also include spatial rainfall and thus dependence of area.

The paper is generally well-written and understandable. Several things are indeed interesting – especially the pixel-area relations depending on rainfall duration as well as the significant climatological differences in the study area.

Thank you very much.

1) The novelty of the paper should be emphasized more in the introduction. I guess compared to former efforts, the novelty here is the application of SMEV also on the areal component rather than single pixels?

Author response: Thank you for the suggestion. Yes, this is the main novelty. The follow has been added to the paper to emphasise this:

*"In this study we apply the SMEV framework to examine extreme precipitation at various spatial scales for the first time, in order to investigate the impact of area size on local extremes. As stated, while the SMEV framework has demonstrated efficacy in successfully estimating extreme rainfall, its prior applications have been confined to either the point (in the case of rain gauge data) or pixel (when utilising radar rainfall data) scale analyses at different temporal scales. We here extend the application of the SMEV to estimate extreme return levels up to 100 years across multiple spatial and temporal scales."* [line 99/107]

2) In my view, it is not clear what the differences between MEV and SMEV are. The authors refer to past publications, but could be relevant to describe the SMEV in a bit more detail here in order to understand how it differs from other methods of extreme value statistics. For example line 157-158 could be detailed further.

Author response: Thank you for the suggestion. The text has been revised and now reads:

*"Extreme precipitation return levels are estimated across the study area using the novel non-asymptotic SMEV framework proposed by Marra et al. (2019a; 2020), a*

*simplified version of the original MEV framework proposed by Marani and Ignaccolo (2015). The MEV and SMEV approaches are based on the concept of 'ordinary events', which are all the independent realisations of the process of interest. Unlike classic extreme value theory, which only exploit a small subset of the data, i.e. the annual maxima or the peaks exceeding a high threshold, they make use of a greater proportion of observations to fit the distribution parameters, thus decreasing the parameter estimation uncertainty. The SMEV is a modified version of the MEV; it neglects the interannual variability of the distribution of ordinary events and in their number of yearly occurrences (Marra et al., 2019a). The SMEV formulation significantly reduces the number of parameters and allows for a direct interpretation of their meaning. This results in a simpler formulation for the non-exceedance probabilities of extreme rainfall, and more robust parameter estimation. Several studies have applied the SMEV to precipitation frequency analysis over different regions (Marra et al., 2020, 2019a; Miniussi and Marra, 2021; Araujo et al., 2023), including over the study area (Marra et al., 2022), and have demonstrated the robustness of the method's assumptions and its ability to reproduce extreme frequencies from relatively short records. The SMEV is used here to estimate precipitation events of varying sizes and durations, so that spatial and temporal effects on extreme precipitation can be analysed"*
[line 157/169]

3) The equation in line 128: Shouldn't it be Z=316R^1.5?

Author response: Yes, thank you, this has been corrected

4) Do you think the discrepancy between gauge and radar is due to differences in climatology, bias adjustment, or radar artifacts? Consulting figure 4 it seems that there are some radar issues close to the radar – maybe something related to scanning and CAPPI generation?

Author response: In addition to the 24-hour radar data derived results, which show this discrepancy with the gauge derived results, we have now calculated the return levels for the 6 validation sites, using radar rainfall data converted to daily timesteps, from 06:00 to 06:00 UTC. This was performed to better match the daily rain gauge data used for the comparison, which is measured from 06:00 to 06:00 UTC. The original 24 h radar data results therefore utilise the exact time window maximising precipitation intensity for each storm, while the 06:00 to 06:00 UTC results select the maximal 06:00 to 06:00 period. The new results have been added to the paper and are also displayed below.

As can be seen, the 06:00 to 06:00 UTC do not exhibit this overestimation compared to the gauge derived results in sites e and f. We posit the following explanation for this in the paper:

*"An analysis by Marra et al. (2022) offers insight into this discrepancy, by examining the time of the day at which the highest short-duration intensities (i.e., the ordinary events in the distribution tail, as in this study defined as the largest 45 %) occur over the study area. They found that the highest offshore intensities*

*tend to occur in the early morning (02:00–08:00 UTC) or morning (08:00–14:00 UTC), and then shift to mostly morning (08:00–14:00 UTC) at the coastline and near inland. This is caused by the convergence created by the superposition of the westerly winds typical of Mediterranean cyclones with land breeze, which is expected to peak in the early morning hours when the sea is the warmest compared to the land. Although Marra et al (2022) focus only on short duration rainfall, and the results are given here for 24 h events, these findings may still explain the discrepancy between the results. Given that sites e and f are situated on the coastline, high rainfall intensities occurring more often in the early mornings between 02:00–08:00 UTC, this could lead to large differences between the maximal 24-hour value and the maximal 06:00 to 06:00 UTC event values. Site d is somewhat more east, with Marra et al. (2022) indicating peak rainfall between 06:00 and 08:00 UTC for this site, whilst sites b and c are the most inshore and present high rainfall intensity peak times of approximately 11:00 – 14:00 UTC and 08:00 – 11:00 UTC respectively. Therefore limiting the daily data to 06:00 to 06:00 UTC may have a lesser impact on these inland sites."* [Line 314/334]

The area of low values around the radar station is caused by the close proximity to the radar station and this is stated in the text:

*"The area of very low values, attributed to data quality issues around the radar location, is clearly visible."* [Line 334/370]

[Figure]

Comparison of the 100 km2, 24 h precipitation intensity return levels derived from 24 h radar data, daily 06:00 to 06:00 UTC data and from rain gauge data.

It should be noted also that there is no CAPPI generation via interpolation or similar in this radar archive.

5) Figure 5. I think it would be very interesting also to include the rain gauges statistics in this figure – and in principle also in figures 6-7. Typically, you would see an underestimation of the radar estimates at short durations. See e.g. Schleiss et al (2020) or Andersen et al 2021. The point scale is indeed interesting to study in addition to the study of the difference between pixel scale and areas of 10, 100 and 500 km2. In regards to point 1 in the conclusion the actual comparison in the manuscript is not made on point scale but on a pixel scale. Even going from point to pixel scale will result in scaling – which will be more dominant for shorter durations than larger ones.

Author response: Thank you for this comment. Actually, radar return levels are adjusted to exactly match the rain gauge ones at the pixel scale (see Marra et al., 2022). This means that the effects of areal mismatch between rain gauges and radar cannot be seen in the estimated return levels. The referee is kindly referred to Marra et al. 2022 for an analysis and discussion of the mismatch between weather radar at the pixel scale and rain gauges (see Fig. 2 in the mentioned paper and related discussion). As this was already examined in the previous study, we keep it out of the objectives of this study.

Thank you for raising the terminology mismatch, the "point" has been corrected to "pixel" throughout the text. Although we note that, as mentioned above, after adjustment the pixel scale is representative of the rain gauge (i.e., point) scale.

6) It would be relevant to present fitted parameter values of shape, scale, kappa, lambda in order to study how the vary over different durations and area. Maybe present selected values in a table. Potentially an empirical relation between model parameter estimates and duration and area could be sought.

Author response: Thank you for the suggestion, which is in line also with Reviewer 1 comment. We have therefore added a figure (rather than table) showing the shape and scale parameters (see figure below) along with the following text:

*"The calculated shape and scale parameters, after correction factors have been applied, are presented in Fig. S4. The effect of both duration and area is clearly visible: the scale parameter decreases with increasing duration and increasing area, with the values converging at long durations – mirroring the behaviour of the return levels presented in Fig. 5. Unlike the scale parameter, the values of the shape parameter do not become more similar for long durations. The parameter displays non-monotonic behaviour, with generally minimal change for durations between 10 min and 1 h, and decreasing for durations between 1 and 6 h (implying an increasing tail heaviness). Very low parameters, between 0.4 and 0.75 (indicating heavy tails), are observed for area sizes greater than the pixel scale, especially over the desert and mountains, while*

*exponential tales (i.e. values close to 1) are observed for the pixel scale."* [Line 454/500]

[Figure]

*Shape and scale parameters (after correction factors have been applied) as a function of area and duration estimated for the desert, coast, and mountains. Shaded areas represent the 90 % confidence interval from 100 bootstrap samples.*